# 🎩 Beyond Magic Words: Sharpness-Aware Prompt Evolving for Robust Large Language Models with TARE

**Guancheng Wan**[1][*] **Lucheng Fu**[2][*] **Haoxin Liu**[2], **Yiqiao Jin**[2], **Hui Yi Leong**[3], **Eric Hanchen Jiang**[1],
**Hejia Geng**[4], **Jinhe Bi**[5], **Yunpu Ma**[5], **Xiangru Tang**[6][†], **B. Aditya Prakash**[2][†], **Yizhou Sun**[1][†], **Wei Wang**[1][†]
[1]University of California, Los Angeles, [2]Georgia Institute of Technology,
[3]University of Chicago, [4]University of Oxford, [5]Ludwig-Maximilians-Universität München, [6]Yale University
`gcwan03@ucla.edu, luchengfu@gatech.edu`

## Abstract

The performance of Large Language Models (LLMs) hinges on carefully engineered prompts. However, prevailing prompt optimization methods, ranging from heuristic edits and reinforcement learning to evolutionary search, primarily target point-wise accuracy. They seldom enforce paraphrase invariance or searching stability, and therefore cannot remedy this brittleness in practice. Automated prompt search remains brittle: small, semantically preserving paraphrases often cause large performance swings. We identify this brittleness as the **textual sharpness** of the **prompt landscape**. In this work, we provide the first formal treatment of textual sharpness in the discrete, semantic space of prompts, together with an operational robustness criterion over a semantic neighborhood; the design is black-box or API-only, requiring no gradients to update the model's parameters. Then we introduce `TARE` (Textual Sharpness-Aware Evolving), a derivative-free framework that alternates between an inner, sampling-based adversarial search that stresses a prompt with hard paraphrases and an outer, robust selection that prefers candidates whose neighborhoods remain strong. We further propose `ATARE`, which learns anisotropic weights to shape the semantic neighborhood and adapts its radius over time to balance exploration and fidelity. Diverse tasks evaluate our methods, whose design for minimizing textual sharpness gap leads to prompts that preserve accuracy under paraphrasing, outperforming accuracy-only prompt search while remaining computationally practical. The code is available at `https://github.com/GuanchengWan/TARE`.

## 1 Introduction

Large Language Models (LLMs) have demonstrated remarkable capabilities across a wide array of natural language understanding and generation tasks (Brown et al., 2020; OpenAI, 2023). The efficacy of these models, however, is critically dependent on the quality of their input prompts. In this vein, prompt engineering aims at manually or automatically discovering optimal prompt structures to guide LLMs toward desired outputs. While automated methods like EvoPrompt and APE (Guo et al., 2023; Zhou et al., 2023) have shown promise, they often produce prompts that are highly sensitive to minor, semantically-equivalent perturbations. This vulnerability largely stems from their primary focus on maximizing point-wise accuracy, which often leads to overfitting on specific phrasings while neglecting stability across the prompt's semantic neighborhood. An optimized prompt that performs well on a given set of inputs may fail dramatically when faced with slight paraphrasing or rephrasing, a phenomenon we term the "sharpness" of the prompt landscape. This brittleness severely limits the real-world reliability and robustness of LLM-based systems.

---

[*]Equal contribution
[†]Corresponding authors

The concept of "sharpness" in optimization landscapes is well-studied in the domain of deep neural networks. It has been shown that models converging to flat minima in the loss landscape exhibit superior generalization performance (Hochreiter & Schmidhuber, 1997). Sharpness-Aware Minimization (SAM) (Foret et al., 2021) and its variants have emerged as powerful techniques to explicitly search these flat minima, thereby improving model robustness and generalization. These methods work by minimizing the loss in a "neighborhood" around the current parameters, effectively smoothing the loss landscape. However, the principles of SAM have been predominantly applied to continuous parameter spaces, such as model weights. Their application to the discrete and combinatorial nature of text-based prompts remains a significant and unexplored challenge.

Therefore, a natural research question arises: **I)** *How can we formally define and quantify the concept of a "sharpness neighborhood" within the discrete, semantic space of textual prompts?* Due to the discrete and semantically rich nature of text, traditional notions of local perturbations (e.g., infinitesimal gradient steps) are fundamentally inapplicable. Instead, we must construct neighborhoods that capture semantic similarity—accounting for paraphrasing and rephrasing—so that local sharpness reflects true linguistic and behavioral proximity relevant to LLMs. Defining such neighborhoods and associated metrics is crucial for robust optimization. Building on this, a closely related question arises: **II)** *How can we design a practical optimization algorithm that navigates this discrete landscape to discover prompts that are both effective and robust to semantic perturbations?* This requires an algorithm that can efficiently explore the prompt space while incorporating a measure of *landscape flatness* into its search process.

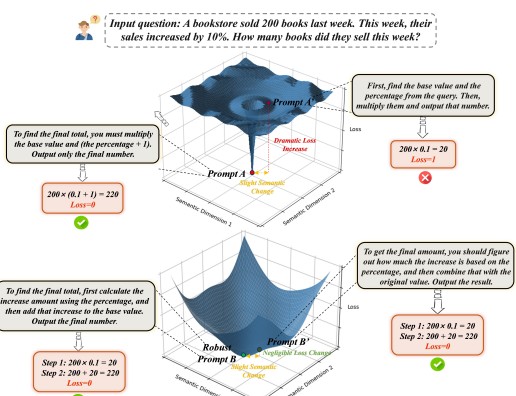

Figure 1: **Problem Illustration.** We illustrate the core challenge in prompt optimization: **I)** conventional methods often find brittle, sharp minima (left), where a **slight semantic change** from an optimal prompt Prompt A to a paraphrase Prompt A' results in a **significant loss increase**. **II)** Our goal is to instead seek flat, stable solutions (right), where a similar **slight semantic change** from a robust prompt Prompt B to its paraphrase Prompt B' only causes the loss to **remain nearly unchanged**, indicating high robustness.

To address these challenges, we introduce *Textual Sharpness-Aware Evolving* (`TARE`), a novel framework inspired by SAM that adapts its core principles for the discrete domain of prompt engineering. Our core contribution is a textual sharpness metric that quantifies prompt robustness by evaluating its performance over a neighborhood of semantically similar variants. We then propose an evolutionary optimization algorithm that iteratively refines prompts, selecting candidates that exhibit both high performance and low sharpness. Furthermore, we introduce an adaptive version, *Adaptive Textual Sharpness-Aware Evolving* (`ATARE`), which dynamically adjusts the neighborhood size during optimization for efficiency and effectiveness. Our contributions are threefold:

❶ **Formalizing Textual Sharpness.** We introduce the first definition of sharpness tailored for the discrete, semantic space of prompts. This is accompanied by a metric to quantify prompt robustness by evaluating performance stability across a semantically coherent neighborhood, bridging the gap between continuous optimization theory and discrete language-based optimization.

❷ **Sharpness-Aware Prompt Evolution.** We propose `TARE`, a novel algorithm designed to explicitly navigate the discrete prompt landscape. By integrating our textual sharpness metric directly into its fitness function, `TARE` effectively co-optimizes for both high task performance and low sharpness, yielding prompts that are both effective and robust. We further enhance this with an adaptive variant, `ATARE`, which dynamically adjusts the neighborhood radius for greater efficiency.

❸ **Superior Robustness and Generalization.** Through extensive experiments on multiple benchmarks, we provide strong empirical evidence that our proposed methods, `TARE` and `ATARE`, consistently discover prompts that are significantly more robust and generalize better to unseen data compared to existing state-of-the-art prompt optimization techniques.

## 2 RELATED WORK

### 2.1 LARGE LANGUAGE MODELS

Large Language Models (LLMs) have rapidly advanced in scale and capability, from early few-shot systems such as GPT-3 (Brown et al., 2020) and GPT-4 (OpenAI, 2023) to general-purpose foundation and open models including PaLM (Chowdhery et al., 2022), Llama 2 (Touvron et al., 2023), Llama 3 (Grattafiori et al., 2024), Mistral 7B (Jiang et al., 2023), and Mixtral (Jiang et al., 2024). Instruction finetuning and alignment further steer model behavior toward user intents (Ouyang et al., 2022; Rafailov et al., 2024; Leong & Wu, 2024; Tang et al., 2025; Wang et al., 2025; 2026). Our problem setting assumes black-box (API-only) access to an LLM (and optionally an evaluator), and focuses on optimizing prompts rather than modifying model parameters, making our approach complementary to parameter-finetuning and alignment.

### 2.2 PROMPT OPTIMIZATION

Prompt engineering has evolved from manual design to automated optimization. Early automated approaches include gradient-free token editing and trigger search (AutoPrompt) (Shin et al., 2020), reinforcement-learning-based optimization (RLPrompt) (Deng et al., 2022), and search-based schemes such as APO (Pryzant et al., 2023). Recent work connects evolutionary algorithms with LLMs or leverages LLMs as optimizers to iteratively propose and select candidates (Guo et al., 2023; Zhou et al., 2023; Fernando et al., 2023; Oh et al., 2024); programmatic frameworks like DSPy compile declarative pipelines into self-improving prompt graphs (Khattab et al., 2023). In parallel, instruction induction and self-instruction curate high-coverage supervision for prompt/-task design (Honovich et al., 2022; Wang et al., 2023b); and reasoning-oriented prompting (CoT, self-consistency, ToT, ReAct, PoT) improves average-case reasoning performance (Wei et al., 2023; Wang et al., 2023a; Yao et al., 2023b;a; Chen et al., 2023). These advanced prompting strategies increasingly allow general-purpose LLMs to perform complex reasoning over temporal events and abstract relational knowledge, domains that have traditionally been dominated by specialized graph-based architectures (Liang et al., 2023; 2025). Nevertheless, most of these methods primarily optimize point-wise metrics on static validation sets and seldom enforce robustness under semantically preserving paraphrases. Our work explicitly targets this failure mode by formalizing textual sharpness in semantic prompt space and optimizing worst-case performance over a neighborhood.

### 2.3 SHARPNESS-AWARE MINIMIZATION

Generalization and robustness in deep learning have been linked to the geometry of the loss landscape, where flat minima often correlate with better generalization (Hochreiter & Schmidhuber, 1997; Keskar et al., 2017). Sharpness-Aware Minimization (SAM) biases solutions toward flatter regions by minimizing loss under worst-case local perturbations (Foret et al., 2021). Subsequent variants extend this idea with scale-invariant updates (ASAM) (Kwon et al., 2021), efficiency-focused or surrogate-gap formulations (Du et al., 2022; Zhuang et al., 2022; Liu et al., 2022b), and friendly/trustworthy adaptations (Li et al., 2024). Related techniques encourage wide valleys via entropy or averaging (Chaudhari et al., 2017; Izmailov et al., 2019) and adversarial weight perturbations (Wu et al., 2020). Unlike these methods that operate in continuous parameter space with gradient access, we instantiate an *analogous* principle in discrete text: we define and measure sharpness over a semantic neighborhood of prompts and develop a black-box, derivative-free algorithm that co-optimizes task performance and local flatness.

## 3 PROBLEM FORMULATION

### 3.1 PROBLEM SETUP AND NOTATION

We consider a black-box large language model and a discrete semantic space of textual prompts. For a supervised task with a training set, the empirical prompt risk is

$$\mathcal{L}_{\mathcal{D}}(p) = \frac{1}{|\mathcal{D}|} \sum_{(x,y) \in \mathcal{D}} \ell\big(\mathcal{M}(p, x), y\big). \tag{1}$$

When the task is generative or judgment based, an evaluator maps model outputs to a numeric loss

$$\ell\big(\mathcal{M}(p,x),y\big) \equiv \mathcal{E}\big(\mathcal{M}(p,x),y\big). \tag{2}$$

The same definitions apply for validation and test. We focus on robust optimization that accounts for semantic neighborhoods of a prompt.

## 3.2 SEMANTIC NEIGHBORHOODS OF PROMPTS

To make this precise, we endow the prompt space with a semantic dissimilarity measure and define the isotropic neighborhood as

$$B\big(p,\rho_{\text{text}}\big) := \big\{\, p' \in \mathcal{P} : d_{\text{text}}(p,p') \le \rho_{\text{text}} \,\big\}. \tag{3}$$

In practice, the semantic dissimilarity $d_{\text{text}}(p,p')$ is defined not as a scalar vector distance, but as a semantic judgment performed by a high-capability LLM.

**Anisotropic neighborhoods.** To capture heterogeneous sensitivity across semantic components of a prompt, we use an anisotropic metric

$$d_{\text{ani},\boldsymbol{W}_p}(p,p') := \big\|\, \boldsymbol{W}_p\, \Delta(p,p') \,\big\|_2, \tag{4}$$

and the corresponding ellipsoidal neighborhood

$$B_p\big(p,\rho_{\text{text}}\big) := \big\{\, p' \in \mathcal{P} : d_{\text{ani},\boldsymbol{W}_p}(p,p') \le \rho_{\text{text}} \,\big\}. \tag{5}$$

## 3.3 TEXTUAL SHARPNESS AND ROBUST RISK

Building on these neighborhoods, the textual sharpness-aware loss is defined as the local worst-case risk over a semantic neighborhood

$$\mathcal{L}_{\text{S}}(p,\rho_{\text{text}}) := \max_{p' \in B(p,\rho_{\text{text}})} \mathcal{L}_{\mathcal{D}}(p'). \tag{6}$$

The corresponding robust optimization problem is

$$\min_{p \in \mathcal{P}} \mathcal{L}_{\text{S}}(p,\rho_{\text{text}}). \tag{7}$$

This mirrors the classical SAM perspective by replacing perturbations in parameter space with textual perturbations in a semantic neighborhood. In the discrete prompt setting, neighborhood exploration during the inner maximization can be operationalized by treating a generator as a sampler.

## 4 METHODOLOGY

### 4.1 OVERVIEW

**Motivation.** Accuracy-only prompt search is brittle: small paraphrases or rephrasings can flip outcomes, exposing sharp, non-flat regions of the prompt landscape discussed in the introduction. Our aim is to explicitly prefer prompts that remain effective under semantically-preserving perturbations. We operationalize the textual sharpness formalization in Sec. 3 into a robust criterion that penalizes local fragility, so that selected prompts demonstrate stability across their semantic neighborhoods. Equivalently, we aim to shrink the textual sharpness gap $\text{Sharp}_{\rho_{\text{text}}}(p)$ defined in Sec. 3.

**Design principles.** (i) Black-box, derivative-free optimization compatible with LLM APIs and evaluator oracles; (ii) semantic neighborhoods that preserve task intent while revealing local sharpness; (iii) an inner worst-case search to expose adversarial neighbors; (iv) an outer robust update that chooses candidates improving the max-risk estimate; and (v) lightweight schedules (radius and budget) for stability under limited compute.

Building on Sec. 3, our goal is to minimize the textual sharpness-aware risk by solving

$$\min_{p \in \mathcal{P}} \max_{p' \in B(p,\rho_{\text{text}})} \mathcal{L}(p'), \tag{8}$$

via a derivative-free, LLM-driven procedure. In this discrete, black-box setting, we rely on sampling-based inner maximization and validation-driven outer selection. We describe `TARE` (isotropic) and `ATARE` (anisotropic) variants that iteratively (i) search adversarial neighbors and (ii) update the prompt to reduce the robust objective.

**Alignment to research questions.** The neighborhood-based objective instantiates **Q1** by defining sharpness in semantic prompt space, while our two-stage, derivative-free robust evolution ad-

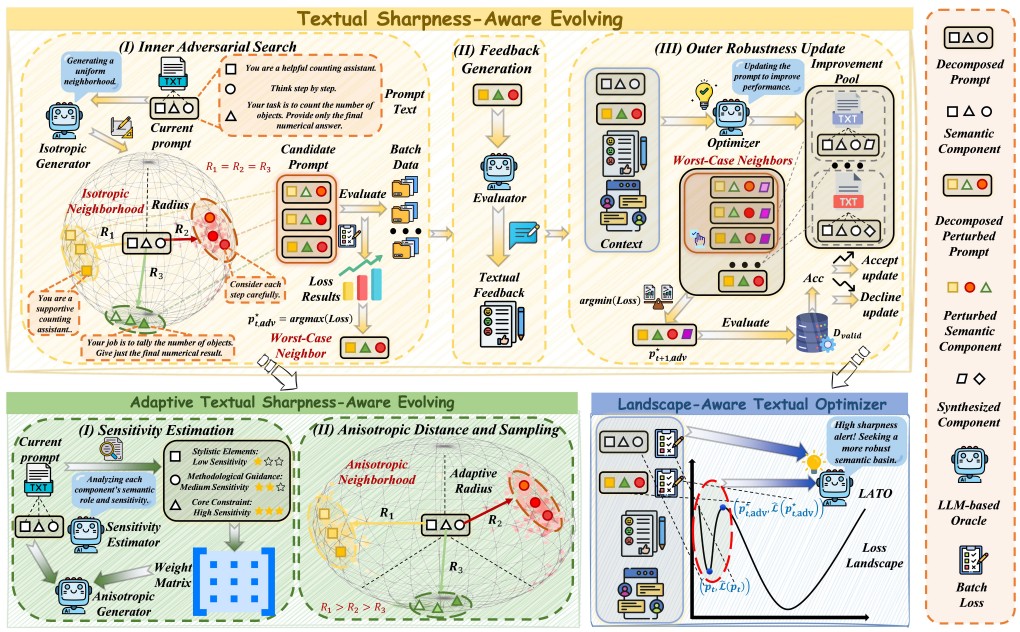

Figure 2: An illustration of our proposed `TARE` framework. (a) **The top panel** shows the main `TARE` loop, consisting of an Inner Adversarial Search, Feedback Generation, and an Outer Robustness Update. (b) **The bottom-left panel** details the `ATARE` mechanism, which uses Sensitivity Estimation to guide an efficient Anisotropic search. (c) **The bottom-right panel** presents the LATO, which perceives the local sharpness to guide updates towards a flatter semantic basin.

dresses **Q2** by providing a practical algorithm that co-optimizes task accuracy and local flatness under semantically-preserving perturbations.

## 4.2 `TARE`: TEXTUAL SHARPNESS-AWARE EVOLVING

Let $p_0 \in \mathcal{P}$ be an initial prompt. At iteration $t = 0, 1, \ldots, T-1$, with radius $\rho_t > 0$ and minibatch $\mathcal{B}_t \subset \mathcal{D}$: To make our algorithm concrete, we trace a running example for a simple text-based object counting task. Let the initial prompt $p_0$ be:

> **Initial Prompt**: *You are a helpful counting assistant. Your task is to count the number of objects. Think step by step and then provide only the final numerical answer.*

For a given input text, the desired output is a single integer (e.g., "3"), and the loss function $\mathcal{L}$ is 1 if the output deviates from this format and 0 otherwise.

**Inner adversarial search.** We sample a candidate set inside the isotropic neighborhood using the generator oracle $\mathcal{G}$:

$$\mathcal{C}_{K_t}(p_t) := \{ p'_1, \ldots, p'_{K_t} \} \sim \mathrm{Sample}\big(\mathcal{G}, p_t, \rho_t, K_t\big), \qquad p'_k \in B(p_t, \rho_t). \tag{9}$$

We evaluate the empirical loss on $\mathcal{B}_t$ and pick the worst case

$$p^\star_{t,\mathrm{adv}} := \operatorname*{arg\,max}_{p' \in \mathcal{C}_{K_t}(p_t)} \widehat{\mathcal{L}}(p'; \mathcal{B}_t), \qquad \widehat{\mathcal{L}}_\mathrm{S}(p_t; \rho_t) := \max_{p' \in \mathcal{C}_{K_t}(p_t)} \widehat{\mathcal{L}}(p'; \mathcal{B}_t). \tag{10}$$

To illustrate this process, consider the isotropic generator $\mathcal{G}$ acting on our initial counting prompt $p_t$. The generator produces perturbations that are semantically preserving, ensuring all candidates $p'$ remain within the defined neighborhood $B(p_t, \rho_t)$. The goal is to test for fragility without altering the fundamental task. For example, the set of candidates $\mathcal{C}_{K_t}(p_t)$ might include:

> ❶ **Candidate 1**: *You are a supportive counting assistant. Your job is to tally the number of objects. Consider each step carefully and then give just the final numerical result.*
>
> ❷ **Candidate 2**: *You are a helpful assistant for counting. Your role is to determine the number of objects. Think through each step and then offer only the final number.*
>
> ❸ **Candidate 3**: *You are a useful assistant for counting objects. Your task is to calculate how many objects there are. Reflect on each step and then present only the final numerical answer.*

A close analysis reveals that while the phrasing, synonyms (*"helpful"* → *"supportive"*, *"count"* → *"tally"*), and sentence structure are altered, the four foundational components of the prompt—persona, task definition, reasoning process, and output format—remain intact across all variations. The algorithm then proceeds to evaluate these candidates to determine if any of these seemingly innocuous rephrasings leads to a performance degradation, thereby revealing the prompt's local sharpness and identifying the adversarial worst-case $p_{t,\text{adv}}^\star$.

**Outer robustness update.** Using an optimizer oracle $\mathcal{O}$, we produce an improvement pool conditioned on the current and adversarial prompts with a semantic budget $\delta_t > 0$:

$$\mathcal{U}_{M_t}(p_t) := \text{Propose}\big(\mathcal{O},\, p_t,\, p_{t,\text{adv}}^\star,\, \delta_t,\, M_t\big) = \{\, \tilde{p}^{(1)}, \ldots, \tilde{p}^{(M_t)} \,\}. \tag{11}$$

In essence, the $\text{Propose}$ function is the core step where the optimizer oracle $\mathcal{O}$ suggests $M_t$ potential improvements by analyzing both the current prompt $p_t$ and its worst-performing neighbor $p_{t,\text{adv}}^\star$. The semantic budget $\delta_t$ restricts edits to preserve task intent and local coherence. We then select the next prompt by robust validation over the union of current and proposals:

$$p_{t+1} := \underset{p' \in \{p_t\} \cup \mathcal{U}_{M_t}(p_t)}{\arg\min}\; \max_{q \in \mathcal{C}_{\tilde{K}}(p')}\; \widehat{\mathcal{L}}(q; \mathcal{B}_t) \quad \text{with} \quad \mathcal{C}_{\tilde{K}}(p') \sim \text{Sample}\big(\mathcal{G},\, p',\, \rho_t,\, \tilde{K}\big). \tag{12}$$

This greedy selection ensures a non-increasing estimate of the robust risk $\widehat{\mathcal{L}}_S(p_t; \rho_t)$ across iterations when the minimizer is attained. Equivalently, it drives down the minibatch estimate of the textual sharpness gap. **Intuition and relation to SAM:** SAM perturbs weights toward the ascent direction and then takes a descent step optimal under that perturbation. `TARE` mirrors this logic in discrete text: the inner sampling-based maximization uncovers the worst paraphrase within $B(p_t, \rho_t)$, and the outer selection moves $p$ toward regions whose neighborhoods are flatter, co-optimizing task performance and robustness.

**Schedules and acceptance.** Typical schedules include: (i) *radius* annealing $\rho_{t+1} = \gamma\, \rho_t$ with $\gamma \in (0, 1]$ when progress stalls; (ii) *semantic budget* $\delta_t$ constrained to preserve task intent; and (iii) *budgets* $(K_t, M_t, \tilde{K})$ chosen to trade off compute and robustness. An iteration is accepted based on a robust validation criterion that evaluates the generalization performance of the worst-case neighbors. Specifically, the worst neighbors $p_{t,\text{adv}}^\star$ and $p_{t+1,\text{adv}}^\star$ are identified on the previous training batch $\mathcal{B}_{t-1}$ and the current training batch $\mathcal{B}_t$, respectively. Evaluating these worst-case neighbors on a separate validation set is crucial to ensure that any observed robustness is a generalizable property and not merely an artifact of a specific training minibatch. The update is therefore accepted only if the new worst neighbor demonstrates superior performance on $\mathcal{D}_{\text{valid}}$:

$$\widehat{\mathcal{L}}(p_{t+1,\text{adv}}^\star; \mathcal{D}_{\text{valid}}) \leq \widehat{\mathcal{L}}(p_{t,\text{adv}}^\star; \mathcal{D}_{\text{valid}}) - \eta, \tag{13}$$

for tolerance $\eta \geq 0$; otherwise we increase search budgets or reduce $\rho_t$.

**From `TARE` to `ATARE`.** Uniform (isotropic) neighborhoods treat all prompt components equally, yet empirical sensitivity is heterogeneous across a prompt's core constraints, methodological guidance, and stylistic elements. This uniformity is inefficient; an ideal search strategy should apply cautious, fine-grained perturbations to sensitive components where the landscape is steep, while exploring robust components more broadly where the landscape is flatter. To achieve this nuanced exploration, we introduce an adaptive, anisotropic variant that learns component-wise weights to shape the neighborhood accordingly, while jointly adapting the neighborhood size $\rho_t$ when needed.

### 4.3 `ATARE`: ADAPTIVE TEXTUAL SHARPNESS-AWARE EVOLVING

The isotropic ball $B(p, \rho)$ may under/over-explore sensitive components of $p$. `ATARE` adapts an ellipsoidal neighborhood via a diagonal weight matrix $\boldsymbol{W}_{p_t} = \text{diag}(\boldsymbol{w}_t)$, where $\boldsymbol{w}_t \in \mathbb{R}_{\geq 0}^m$ scores component-wise sensitivity.

**Why anisotropy?** Different parts of a prompt contribute unequally to its behavior: core constraints, methodological guidance, and stylistic elements exhibit heterogeneous sensitivity. For instance, in our counting-task example, the persona *"You are a helpful..."* is a stylistic element, the instruction to *"think step by step"* provides methodological guidance, and the rule to *"provide only the final numerical answer"* is a core constraint. An isotropic ball may overshoot sensitive tokens or underexplore robust ones. `ATARE` learns component-wise weights to shape an ellipsoidal neighborhood, applying finer, more constrained perturbations where the landscape is steep, while allowing for

broader exploration in more stable regions. This accelerates convergence and reduces over-editing of fragile components.

**Sensitivity estimation.** Formally, we define the sensitivity $s_{t,j}$ of a prompt component $j$ as the worst-case performance degradation caused by its semantic perturbation. For a neighborhood of candidate prompts, this is expressed as:

$$s_{t,j} := \max_{p' \in \mathcal{C}_t} \mathbb{I}(p' \text{ modifies } j) \cdot \left[ \mathcal{L}(p') - \mathcal{L}(p_t) \right]_+, \tag{14}$$

where $\mathbb{I}(\cdot)$ indicates whether component $j$ is modified. Crucially, empirical observation reveals that this mathematical sensitivity maps directly to semantic roles: modifying *Constraints* (e.g., output formats) typically triggers immediate task failure (high loss spike, resulting in large $s_{t,j}$), whereas varying *Style* induces negligible performance shifts (resulting in small $s_{t,j}$). Leveraging this alignment, we efficiently approximate these sensitivity values by decomposing the prompt into three hierarchical tiers: *Constraint*, *Method*, and *Style*. We then determine the anisotropic weights $w_{t,j}$ as a monotonic function of sensitivity, reflecting the observed hierarchy:

$$w_{t,j} \propto s_{t,j}, \quad \text{where } s_{\text{Constraint}} > s_{\text{Method}} > s_{\text{Style}}. \tag{15}$$

This assignment reflects the inherent sensitivity. For instance, in our object counting task, a minor paraphrase of the output format rule (a *Constraint*) often leads to a high loss due to parsing failures; consequently, our framework assigns a large weight $w_{t,j}$ to this component to strictly limit its perturbation radius.

**Anisotropic distance and sampling.** Using $\boldsymbol{W}_{p_t} = \text{diag}(\boldsymbol{w}_t)$, define $d_{\text{ani}}(p_t, p'; \boldsymbol{W}_{p_t}) = \|\boldsymbol{W}_{p_t} \Delta(p_t, p')\|_2$ and the ellipsoid

$$B_{p_t}(p_t, \rho_t) = \left\{ p' : d_{\text{ani}}(p_t, p'; \boldsymbol{W}_{p_t}) \le \rho_t \right\}. \tag{16}$$

Here, a high weight $w_{t,j}$ for a sensitive component penalizes large edits, thus requiring smaller perturbations to stay within the neighborhood. Candidate generation is therefore biased *away* from sensitive components by sampling edit indices with a probability inversely proportional to their sensitivity:

$$\Pr\{\text{edit component } j\} \propto (1/w_{t,j})^\beta, \qquad \beta \ge 1. \tag{17}$$

This ensures robust components are explored broadly while fragile ones are perturbed cautiously. The inner/outer steps then mirror `TARE` with $B$ replaced by $B_{p_t}$.

This anisotropic sampling process culminates in the generation of complete, holistic prompts where the degree of variation in each component reflects its learned sensitivity. For instance, in our counting-task example's candidates below, the low-sensitivity persona is creatively reimagined—from a *"helpful counting assistant"* to a *"cheerful counter"*. In contrast, the high-sensitivity constraint on the output format is meticulously preserved; although its phrasing is subtly varied (e.g., *"give just the final number"* or *"present only the final digit answer"*), the core directive to output only a number remains unchanged:

> ❶ **Candidate 1**: *You are a friendly counting helper. Your task is to count the objects. Work through the process step by step and then give just the final number.*
>
> ❷ **Candidate 2**: *You are an assistant designed to count things. First reason through the counting carefully, then respond with the single final numeric result.*
>
> ❸ **Candidate 3**: *As a cheerful counter, your role is to determine how many items there are. Go through your reasoning in order, but at the end present only the final digit answer.*

## 4.4 Landscape-Aware Textual Optimizer

The Outer robustness update step relies on an optimizer oracle $\mathcal{O}$ to instantiate the Propose function. We operationalize this oracle with a potent, landscape-aware implementation, which we term the **Landscape-Aware Textual Optimizer (LATO)**. LATO realizes this step as a principled, landscape-guided update, formally defining the Propose function with its full set of inputs:

$$\text{Propose} := \left\{ \mathcal{O}_{\text{LATO}}^{(i)} \left( p_t, p_{t,\text{adv}}^\star, \widehat{\mathcal{L}}(p_t; \mathcal{B}_t), \widehat{\mathcal{L}}(p_{t,\text{adv}}^\star; \mathcal{B}_t), \text{Feedback}\left( \widehat{\mathcal{L}}(p_{t,\text{adv}}^\star; \mathcal{B}_t) \right), \delta_t \right) \right\}_{i=1}^{M_t}. \tag{18}$$

The update mechanism of LATO is designed to enhance robustness directly. Instead of merely correcting errors at its current position $p_t$, LATO analyzes the textual feedback,

Table 1: **Main results across different backbone engines.** We report accuracy (%) and the relative improvement over TextGrad. The best and second-best results are highlighted with **bold** and underline, respectively.

| Dataset | Model | BACKBONE: GPT-4o | | | | | BACKBONE: Claude 3.5 Sonnet | | | | |
|---|---|---|---|---|---|---|---|---|---|---|---|
| | | COT | TEXTGRAD | REVOLVE | TARE | ATARE | COT | TEXTGRAD | REVOLVE | TARE | ATARE |
| Object Counting | GPT-3.5 | $77.9_{\downarrow 10.1}$ | $88.0$ | $89.8_{\uparrow 1.8}$ | $\underline{90.2}_{\uparrow 2.2}$ | $\mathbf{91.0}_{\uparrow 3.0}$ | $77.9_{\downarrow 5.4}$ | $83.3$ | $87.5_{\uparrow 4.2}$ | $\mathbf{90.4}_{\uparrow 7.1}$ | $\underline{88.0}_{\uparrow 4.7}$ |
| | Gemini 1.5 Flash 8B | $82.0_{\downarrow 1.3}$ | $83.3$ | $83.5_{\uparrow 0.2}$ | $\underline{84.7}_{\uparrow 1.4}$ | $\mathbf{85.7}_{\uparrow 2.4}$ | $82.0_{\downarrow 6.5}$ | $88.5$ | $90.0_{\uparrow 1.5}$ | $\mathbf{94.4}_{\uparrow 5.9}$ | $91.0_{\uparrow 2.5}$ |
| | Gemini 1.5 Pro | $94.0_{\uparrow 0.0}$ | $94.0$ | $94.0_{\uparrow 0.0}$ | $\mathbf{97.3}_{\uparrow 3.3}$ | $\mathbf{97.3}_{\uparrow 3.3}$ | $94.0_{\downarrow 3.0}$ | $97.0$ | $97.7_{\uparrow 0.7}$ | $\underline{98.0}_{\uparrow 1.0}$ | $\mathbf{98.3}_{\uparrow 1.3}$ |
| | Llama 3.1 8B Instruct | $86.0_{\downarrow 2.6}$ | $88.6$ | $88.2_{\downarrow 0.4}$ | $\underline{91.0}_{\uparrow 2.4}$ | $\mathbf{92.4}_{\uparrow 3.8}$ | $86.0_{\downarrow 3.5}$ | $89.5$ | $88.0_{\downarrow 1.5}$ | $90.6_{\uparrow 1.1}$ | $\mathbf{93.5}_{\uparrow 4.0}$ |
| | Llama 3 8B Instruct | $80.0_{\downarrow 5.8}$ | $85.8$ | $86.8_{\uparrow 1.0}$ | $\underline{88.7}_{\uparrow 2.9}$ | $\mathbf{90.3}_{\uparrow 4.5}$ | $80.0_{\downarrow 2.0}$ | $82.0$ | $84.3_{\uparrow 2.3}$ | $\underline{89.5}_{\uparrow 7.5}$ | $\mathbf{93.6}_{\uparrow 11.6}$ |
| Temporal Sequences | GPT-3.5 | $79.0_{\downarrow 2.0}$ | $81.0$ | $84.0_{\uparrow 3.0}$ | $\underline{87.5}_{\uparrow 6.5}$ | $\mathbf{88.0}_{\uparrow 7.0}$ | $79.0_{\downarrow 7.7}$ | $86.7$ | $84.4_{\downarrow 2.3}$ | $\underline{88.0}_{\uparrow 1.3}$ | $\mathbf{89.0}_{\uparrow 2.3}$ |
| | Gemini 1.5 Flash 8B | $92.0_{\downarrow 0.5}$ | $92.5$ | $93.0_{\uparrow 0.5}$ | $\underline{94.3}_{\uparrow 1.8}$ | $\mathbf{95.2}_{\uparrow 2.7}$ | $92.0_{\downarrow 2.0}$ | $94.0$ | $94.7_{\uparrow 0.7}$ | $\underline{95.3}_{\uparrow 1.3}$ | $\mathbf{95.7}_{\uparrow 1.7}$ |
| | Gemini 1.5 Pro | $96.0_{\downarrow 1.7}$ | $97.7$ | $97.7_{\uparrow 0.0}$ | $\mathbf{98.0}_{\uparrow 0.3}$ | $\mathbf{98.0}_{\uparrow 0.3}$ | $96.0_{\downarrow 1.0}$ | $97.0$ | $98.0_{\uparrow 1.0}$ | $\underline{98.5}_{\uparrow 1.5}$ | $\mathbf{98.8}_{\uparrow 1.8}$ |
| | Llama 3.1 8B Instruct | $86.0_{\downarrow 2.3}$ | $88.3$ | $88.3_{\uparrow 0.0}$ | $\underline{90.0}_{\uparrow 1.7}$ | $\mathbf{91.0}_{\uparrow 2.7}$ | $86.0_{\downarrow 7.0}$ | $93.0$ | $89.6_{\downarrow 3.4}$ | $\underline{93.7}_{\uparrow 0.7}$ | $\mathbf{94.3}_{\uparrow 1.3}$ |
| | Llama 3 8B Instruct | $84.0_{\uparrow 0.0}$ | $84.0$ | $84.5_{\uparrow 0.5}$ | $\underline{85.0}_{\uparrow 1.0}$ | $\mathbf{88.7}_{\uparrow 4.7}$ | $84.0_{\downarrow 7.5}$ | $91.5$ | $89.2_{\downarrow 2.3}$ | $\mathbf{94.0}_{\uparrow 2.5}$ | $\mathbf{94.0}_{\uparrow 2.5}$ |
| Tracking Shuffled Objects | GPT-3.5 | $62.0_{\downarrow 4.3}$ | $66.3$ | $65.7_{\downarrow 0.6}$ | $\mathbf{72.0}_{\uparrow 5.7}$ | $69.0_{\uparrow 2.7}$ | $62.0_{\downarrow 13.0}$ | $75.0$ | $72.2_{\downarrow 2.8}$ | $\mathbf{77.0}_{\uparrow 2.0}$ | $\mathbf{77.0}_{\uparrow 2.0}$ |
| | Gemini 1.5 Flash 8B | $82.0_{\downarrow 1.0}$ | $83.0$ | $82.5_{\downarrow 0.5}$ | $\underline{88.6}_{\uparrow 5.6}$ | $\mathbf{93.7}_{\uparrow 10.7}$ | $82.0_{\downarrow 5.3}$ | $87.3$ | $89.8_{\uparrow 2.5}$ | $\underline{91.3}_{\uparrow 4.0}$ | $\mathbf{94.0}_{\uparrow 6.7}$ |
| | Gemini 1.5 Pro | $99.0_{\uparrow 0.0}$ | $99.0$ | $99.0_{\uparrow 0.0}$ | $99.0_{\uparrow 0.0}$ | $99.0_{\uparrow 0.0}$ | $99.0_{\uparrow 0.0}$ | $99.0$ | $99.0_{\uparrow 0.0}$ | $99.0_{\uparrow 0.0}$ | $99.0_{\uparrow 0.0}$ |
| | Llama 3.1 8B Instruct | $82.0_{\downarrow 4.3}$ | $86.3$ | $83.7_{\downarrow 2.6}$ | $\underline{90.0}_{\uparrow 3.7}$ | $\mathbf{93.5}_{\uparrow 7.2}$ | $82.0_{\uparrow 0.8}$ | $81.2$ | $79.2_{\downarrow 2.0}$ | $\underline{91.2}_{\uparrow 10.0}$ | $\mathbf{93.0}_{\uparrow 11.8}$ |
| | Llama 3 8B Instruct | $50.0_{\downarrow 5.5}$ | $55.5$ | $52.7_{\downarrow 2.8}$ | $\underline{57.5}_{\uparrow 2.0}$ | $\mathbf{67.7}_{\uparrow 12.2}$ | $50.0_{\downarrow 14.5}$ | $64.5$ | $66.8_{\uparrow 2.3}$ | $\underline{72.3}_{\uparrow 7.8}$ | $\mathbf{78.5}_{\uparrow 14.0}$ |
| GSM8K | GPT-3.5 | $72.9_{\downarrow 8.0}$ | $80.9$ | $82.1_{\uparrow 1.2}$ | $\mathbf{83.0}_{\uparrow 2.1}$ | $\underline{82.3}_{\uparrow 1.4}$ | $72.9_{\downarrow 8.2}$ | $81.1$ | $80.1_{\downarrow 1.0}$ | $\mathbf{83.7}_{\uparrow 2.6}$ | $\mathbf{83.7}_{\uparrow 2.6}$ |
| | Gemini 1.5 Flash 8B | $88.6_{\downarrow 1.0}$ | $89.6$ | $89.4_{\downarrow 0.2}$ | $\mathbf{90.1}_{\uparrow 0.5}$ | $\underline{89.7}_{\uparrow 0.1}$ | $88.6_{\downarrow 0.1}$ | $88.7$ | $88.9_{\uparrow 0.2}$ | $\underline{89.6}_{\uparrow 0.9}$ | $\mathbf{89.7}_{\uparrow 1.0}$ |
| | Gemini 1.5 Pro | $92.9_{\downarrow 0.4}$ | $93.3$ | $93.0_{\downarrow 0.3}$ | $\mathbf{95.5}_{\uparrow 2.2}$ | $\underline{94.7}_{\uparrow 1.4}$ | $92.9_{\downarrow 2.4}$ | $95.3$ | $95.3_{\uparrow 0.0}$ | $\mathbf{96.1}_{\uparrow 0.8}$ | $\underline{95.5}_{\uparrow 0.2}$ |
| | Llama 3.1 8B Instruct | $84.9_{\uparrow 0.0}$ | $84.9$ | $84.9_{\uparrow 0.0}$ | $\underline{86.2}_{\uparrow 1.3}$ | $\mathbf{86.4}_{\uparrow 1.5}$ | $84.9_{\downarrow 1.3}$ | $86.2$ | $86.4_{\uparrow 0.2}$ | $\underline{86.9}_{\uparrow 0.7}$ | $\mathbf{87.7}_{\uparrow 1.5}$ |
| | Llama 3 8B Instruct | $81.8_{\uparrow 0.0}$ | $81.8$ | $81.8_{\uparrow 0.0}$ | $81.8_{\uparrow 0.0}$ | $81.8_{\uparrow 0.0}$ | $81.8_{\uparrow 0.0}$ | $81.8$ | $81.8_{\uparrow 0.0}$ | $81.8_{\uparrow 0.0}$ | $81.8_{\uparrow 0.0}$ |

$\text{Feedback}\big(\widehat{\mathcal{L}}(p_{t,\text{adv}}^{\star}; \mathcal{B}_t)\big)$, which is derived from the point of highest local loss. It then applies this insight to refine $p_t$. This process preemptively addresses the sharpest vulnerabilities in the prompt's immediate semantic neighborhood. By learning from the failure modes of its neighbors, the optimizer guides $p_t$ to become inherently more robust against similar types of semantic perturbations in the future.

This approach is powerful because LATO is, by construction, landscape-aware. By processing the two distinct prompt-loss pairs, $(p_t, \widehat{\mathcal{L}}(p_t; \mathcal{B}_t))$ and $(p_{t,\text{adv}}^{\star}, \widehat{\mathcal{L}}(p_{t,\text{adv}}^{\star}; \mathcal{B}_t))$, it directly perceives the local sharpness of the semantic landscape. This awareness of the landscape's geometry—the steepness of the loss increase from $p_t$ to $p_{t,\text{adv}}^{\star}$ and this empirically found worst-case direction—allows LATO to modulate its optimization strategy. It makes more informed decisions about both the direction and magnitude of the required edits, steering the prompt trajectory towards a demonstrably "flatter" and more stable semantic basin.

To illustrate this mechanism concretely, consider an object counting task. A candidate prompt *"Count the items below"* might exhibit high accuracy but fail significantly when paraphrased to other prompts, revealing a sharp, brittle peak (large loss gap). LATO detects this instability and steers optimization away from it. In contrast, a prompt like *"List the items one by one and count them"* maintains high performance across its semantic neighborhood, indicating a robust, flat basin. Unlike standard optimizers that might greedily prefer the former for its brevity or marginal pointwise advantage, LATO leverages this landscape information to correctly converge on the robust solution.

Operationally, LATO is instantiated using a powerful LLM as the core of the optimizer oracle $\mathcal{O}_{\text{LATO}}$. The update process can be expressed as the LLM generating a new prompt based on a structured meta-prompt, $\Pi_{\text{LATO}}$, which contains all the landscape information:

$$\tilde{p}^{(i)} := \text{LLM}\left(\Pi_{\text{LATO}}\left(p_t, p_{t,\text{adv}}^{\star}, \widehat{\mathcal{L}}(p_t; \mathcal{B}_t), \widehat{\mathcal{L}}(p_{t,\text{adv}}^{\star}; \mathcal{B}_t), \text{Feedback}\big(\widehat{\mathcal{L}}(p_{t,\text{adv}}^{\star}; \mathcal{B}_t)\big), \delta_t\right)\right). \quad (19)$$

Here, $\Pi_{\text{LATO}}$ represents a meta-prompt template that synthesizes all the landscape-aware inputs from Equation (18) into a coherent, actionable instruction. The semantic budget $\delta_t$ acts as a crucial constraint, ensuring that the edits proposed by the LLM remain coherent and preserve the core intent of the task. The LLM then executes this instruction to generate an improved candidate prompt

$\tilde{p}^{(i)}$, which forms an element of the proposal set Propose, effectively acting as a reasoning engine that performs a landscape-guided optimization step.

# 5 EXPERIMENTS

We comprehensively evaluate our proposed methods, `TARE` and `ATARE`, through four axes: **Q1** (Superiority), **Q2** (Effectiveness), **Q3** (Resilience), and **Q4** (Sensitivity). The answers of **Q1-Q3** are illustrated in Sec. 5.2-Sec. 5.4, and sensitivity analysis (**Q4**) can be found in the Appendix C.

## 5.1 EXPERIMENTAL SETUP

**Tasks and Datasets.** We evaluate our methods on four challenging reasoning tasks: three from the Big Bench Hard benchmark (Suzgun et al., 2022; Srivastava et al., 2023)—**Object Counting**, **Temporal Sequences**, and **Tracking Shuffled Objects (Five Objects)**—and the **GSM8K** dataset (Cobbe et al., 2021). For evaluation, our primary metric is Accuracy (Acc), measured by a strict string-based exact match on the final numerical answer (Yuksekgonul et al., 2024). Further details on datasets and implementation are provided in Appendix A.

**LLM Backends.** Our experiments are conducted on a diverse set of five LLM backends: **GPT-3.5-turbo-0125**, **Gemini 1.5 Flash 8B**, **Gemini 1.5 Pro**, **Llama 3.1 8B Instruct**, and **Llama 3 8B Instruct**. To ensure a fair and controlled comparison, the optimizer and evaluator oracles for all methods are powered by two universal backbones: **GPT-4o** and **Claude 3.5 Sonnet**.

**Counterparts.** We compare our methods, `TARE` and `ATARE`, against three key baselines: Zero-shot Chain-of-Thought (CoT) (Kojima et al., 2023; Wei et al., 2023), TextGrad (Yuksekgonul et al., 2024), and Revolve (Zhang et al., 2025).

## 5.2 SUPERIORITY

To answer **Q1**, we present the main prompt optimization results in Tab. 1. We summarize our key observations as follows (**Obs.**): **Obs.** ❶ Our proposed methods, `TARE` and `ATARE`, consistently achieve state-of-the-art performance, outperforming all baselines, including TextGrad and Revolve, across nearly all evaluated tasks and LLM backbones. This significant performance gap stems from a fundamental difference in optimization objectives. While baselines are designed to maximize point-wise accuracy, our framework explicitly seeks robust solutions by optimizing for the worst-case performance within a semantic

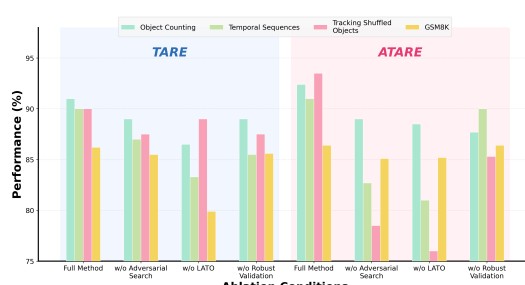

Figure 3: Ablation study of the key components: the Inner Adversarial Search, the LATO, and the Robust Validation. For an in-depth analysis, please refer to Sec. 5.3.

neighborhood, leading to more generalizable and effective prompts. **Obs.** ❷ `ATARE` consistently demonstrates a performance advantage over `TARE` in most scenarios. This underscores the benefit of its adaptive, anisotropic search mechanism, which intelligently perturbs prompt components based on their learned sensitivity. This more nuanced search strategy consistently discovers superior solutions within the prompt landscape. **Obs.** ❸ The framework's superiority shows **broad universality**, with substantial performance gains observed across diverse architectures, including proprietary models like GPT-3.5 and Gemini 1.5 Pro, as well as open-source models like the Llama 3 Instruct series. This confirms that our sharpness-aware approach is a model-agnostic and widely applicable solution for robust prompt optimization.

## 5.3 EFFECTIVENESS

To address **Q2**, we conducted an ablation study on the key mechanisms of our framework using the Llama 3.1 8B Instruct model, with results shown in Figure 3. The chart clearly shows that the full `TARE` and `ATARE` frameworks perform best, and removing any of their core components leads to a significant drop in performance. Specifically, the **Inner Adversarial Search** is essential for

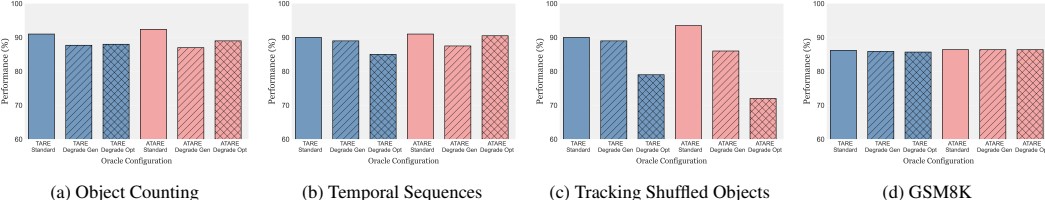

(a) Object Counting    (b) Temporal Sequences    (c) Tracking Shuffled Objects    (d) GSM8K

Figure 4: Resilience analysis of `TARE` and `ATARE` under oracle degradation, where the powerful GPT-4o oracles are replaced with a weaker Llama 3.1 8B model. For an in-depth analysis, please refer to Sec. 5.4.

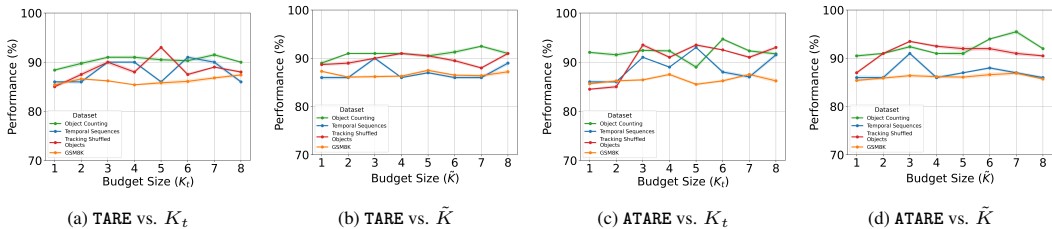

(a) `TARE` vs. $K_t$    (b) `TARE` vs. $\tilde{K}$    (c) `ATARE` vs. $K_t$    (d) `ATARE` vs. $\tilde{K}$

Figure 5: Sensitivity analysis of `TARE` and `ATARE` with respect to the inner adversarial search budget ($K_t$) and the outer robust validation budget ($\tilde{K}$). For an in-depth analysis, please refer to Appendix C.

finding challenging perturbations, the **LATO** optimizer uses landscape information to make smarter updates, and the **Robust Validation** criterion ensures that improvements generalize well. Finally, the consistent superiority of `ATARE` over `TARE` (detailed in Tab. 1) serves as a direct ablation for the **Anisotropic Search**, confirming the benefits of an adaptive strategy. When these components work together, the framework reaches its peak effectiveness, validating our design choices.

## 5.4 RESILIENCE

To assess the resilience of our framework (**Q3**), we evaluate its performance under two challenging conditions: (i) degrading its powerful GPT-4o oracles by separately replacing the **Generator** and the **Optimizer** with a weaker Llama 3.1 8B model, and (ii) drastically reducing the search budgets for the inner adversarial search $K_t$ and outer robust validation $\tilde{K}$. The results, illustrated in Figure 4 and Figure 5, demonstrate the framework's remarkable stability. As shown in Figure 4, even when individual core oracles are weakened, the performance degradation is remarkably graceful. With the exception of the Optimizer degradation on the Tracking Shuffled Objects task, the accuracy drop across all other conditions is consistently maintained within a 5% margin. Similarly, as shown in our sensitivity analysis (Figure 5), when the perturbation budgets ($K_t, \tilde{K}$) are reduced to a minimal value of 1 or 2, the framework's performance remains highly stable, exhibiting only a minor decrease relative to its performance at a budget of 3. This dual resilience proves that our sharpness-aware approach is robust to component degradation and computationally efficient, maintaining strong performance even under such challenging conditions.

## 6 CONCLUSION

Reliable prompt optimization begins with naming the right failure mode. Our work identifies and formalizes the overlooked problem of textual sharpness—the tendency of a prompt to collapse under semantically equivalent paraphrases—and reframes prompt optimization from chasing point-wise accuracy to seeking neighborhood-stable solutions. We instantiate this perspective with `TARE`, a black-box, derivative-free procedure that adversarially probes a semantic neighborhood and selects candidates by their worst-case minibatch performance, and with `ATARE`, which learns anisotropic weights and adaptively schedules the neighborhood radius to balance exploration and fidelity. Both variants are API-only and gradient-free; the adaptive version adds only linear overhead in the number of semantic components while enforcing a fixed-margin decrease per accepted step. Across diverse tasks, `TARE` and `ATARE` consistently reduce the textual sharpness gap and preserve accuracy under paraphrasing, surpassing accuracy-only baselines while remaining computationally practical. Looking ahead, we see opportunities to extend textual sharpness-aware evolution to multi-turn and tool-augmented settings, to design task-aware semantic neighborhoods and edit families, and to deepen theory connecting textual sharpness with generalization in real-world LLM systems.

## REPRODUCIBILITY

To facilitate the reproducibility of our findings, we will release the source code for our `TARE` and `ATARE` framework upon publication, accessible via an anonymous GitHub link. All experimental settings, including key hyperparameters for the optimization process, are detailed in Appendix A.3. All prompts used to generate the experimental data are provided in Appendix E. Our experiments were conducted on a server equipped with 8 NVIDIA GeForce RTX 3090 GPUs.

## LLM USAGE

We acknowledge the use of Google's Gemini 2.5 Pro as a writing assistant in the preparation of this manuscript. Its role was confined to improving the clarity and readability of the text, offering suggestions for grammatical corrections, and refining the structure of captions for figures and tables.

The model's contributions were strictly limited to surface-level text and formatting; it was not used for research ideation, experimental design, implementation, data analysis, or writing the core technical content. All outputs from the model were critically reviewed, edited, and approved by the authors, who bear full responsibility for the final manuscript.

## ETHICS AND SOCIETY IMPACT

The focus of this research is a new methodology for making prompts for LLMs more robust. As a purely algorithmic study, it does not involve human participants, the collection of private data, or direct deployment in sensitive, real-world scenarios. Our contributions are limited to the optimization algorithm itself, and we do not introduce new data that could present risks related to privacy or bias. We acknowledge that more capable language models can have a broad societal impact. However, our work is intended for academic purposes and is demonstrated on established reasoning benchmarks, not on applications involving potential misuse or deception. In summary, this research presents no direct ethical risks and aligns with the principles of creating trustworthy and transparent AI.

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

# A    EXPERIMENTAL DETAILS

## A.1    DATASET DETAILS

To assess the effectiveness of our framework, we conduct experiments on four diverse and challenging reasoning tasks. Consistent with prior work (Yuksekgonul et al., 2024), the evaluation metric is string-based exact match accuracy. A detailed description is provided below:

- **BIG-Bench Hard Tasks** (Suzgun et al., 2022; Srivastava et al., 2023). BIG-Bench Hard is a suite of 23 challenging tasks from the BIG-Bench benchmark, specifically selected because prior language models had failed to outperform the average human-rater on them. These tasks often require multi-step reasoning, making them suitable for evaluating advanced model capabilities. From this benchmark, we select three distinct tasks:
  - **Object Counting:** Given a list of items and their quantities, the task is to determine the total number of items belonging to a specific category.
  - **Temporal Sequences:** Given a series of events and activities a person has completed, the task is to determine a time they might have been free for another activity.
  - **Tracking Shuffled Objects (Five Objects):** Given the initial positions of several objects and a series of pairwise swaps, the task is to determine the final position of each object.

  For the **Object Counting** task, we adopt the data split of 50 training, 100 validation, and 100 test samples from TextGrad (Yuksekgonul et al., 2024). For **Temporal Sequences** and **Tracking Shuffled Objects (Five Objects)**, we follow an identical splitting methodology.
- **GSM8K** (Cobbe et al., 2021). To further assess mathematical reasoning, we use this widely-used benchmark consisting of grade-school math word problems that require multi-step reasoning. For this task, we adopt the dataset splits provided by DSPy (Khattab et al., 2023), which include 200 training, 300 validation, and 1319 test samples.

## A.2    COUNTERPART DETAILS

This section provides an overview of the baseline approaches employed in our study for comparison.

- **Zero-shot Chain-of-Thought (CoT)** (Kojima et al., 2023; Wei et al., 2023). A foundational baseline that elicits multi-step reasoning by prompting the model with instructions like "Think step-by-step" before it provides a final answer.
- **TextGrad** (Yuksekgonul et al., 2024). A first-order optimization method that treats natural language feedback from an evaluator LLM as a "textual gradient" to iteratively refine variables based on immediate, local feedback.
- **Revolve** (Zhang et al., 2025). An optimization method that extends first-order techniques by tracking how system responses evolve across iterations. By incorporating this historical context, Revolve aims for more stable optimization and to escape the local optima that can trap methods relying on single-step feedback.

## A.3    IMPLEMENTATION DETAILS

Our experiments are conducted on a diverse set of five LLM backends: **GPT-3.5-turbo-0125**, **Gemini 1.5 Flash 8B**, **Gemini 1.5 Pro**, **Llama 3.1 8B Instruct**, and **Llama 3 8B Instruct**. To ensure a fair and controlled comparison, our setup relies on a universal backbone engine for three key roles: generators, optimizers, and evaluators. For these backbone roles, we employ two powerful models: **GPT-4o** and **Claude 3.5 Sonnet**.

For all iterative methods, we follow the experimental setup in Revolve (Zhang et al., 2025), using a batch size of 3 across 12 optimization iterations, processing a total of 36 training examples. For our sharpness-aware methods, we set the key search budgets to $K_t = 3$ (inner adversarial search), $M_t = 1$ (proposal pool size), and $\tilde{K} = 3$ (outer robust validation). For LLM generation, our configuration largely mirrors that of Revolve (Zhang et al., 2025): we allow a maximum of 2000 new tokens and use a top-p value of 0.99. To ensure maximum reproducibility, we set the decoding temperature to 0 for all models.

## B    COMPLEXITY AND PRACTICAL NOTES

Beyond asymptotic cost, the design rationale is that the inner loop diagnoses textual sharpness while the outer loop enforces progress on the robust objective; radius annealing preserves semantics, and acceptance tests prevent regressions. The framework is modular: $\mathcal{G}$ (candidate generators), $\mathcal{O}$ (optimizers), and evaluators $\mathcal{E}$ can be swapped without changing the principle. The choice of $\rho_t$ can be guided by paraphrase detection or embedding-similarity thresholds to maintain semantic fidelity.

Each iteration evaluates $K_t$ adversarial and $(M_t + 1)\tilde{K}$ robust losses on a minibatch, totaling $O\big((K_t + (M_t + 1)\tilde{K})|\mathcal{B}_t|\big)$ calls to $\mathcal{M}$ and $\mathcal{E}$. `ATARE` adds an $O(m)$ overhead for weight updates and negligible cost for adaptive radius updates. In practice: (i) reuse evaluations across inner/outer loops; (ii) maintain a replay buffer of high-loss neighbors to warm-start future inner maximizations; and (iii) set $\rho_t$ to preserve semantic intent while revealing sharp regions.

## C    SENSITIVITY

To address **Q4**, we perform a sensitivity analysis on the two key search budget hyperparameters of our framework: the inner adversarial search budget $K_t$ and the outer robust validation budget $\tilde{K}$. As illustrated in Figure 5, we evaluate the performance of `TARE` and `ATARE` on the Llama 3.1 8B model, using GPT-4o as the backbone engine, by systematically varying one budget within the range of [1, 8] while keeping the other fixed at a moderate value of 3. The results indicate that our framework is not highly sensitive to the precise choice of these parameters. Performance generally improves as the budgets increase from 1 to 3 and then stabilizes, exhibiting only minor fluctuations for values up to 8. This demonstrates that our methods can achieve strong, robust performance without requiring extensive hyperparameter tuning, as a relatively small budget (e.g., $K_t = \tilde{K} = 3$) is sufficient to capture the benefits of our sharpness-aware approach.

## D    SOLUTION OPTIMIZATION

While the core of our work focuses on prompt optimization, the principles of our framework can be extended to other complex textual domains. A critical application is **Solution Optimization**, which involves the iterative refinement of multi-step reasoning chains. Unlike prompts, solutions are highly structured and logically interlocked, presenting unique challenges that require a tailored approach.

### D.1    APPLYING `ATARE` TO FRAGILE REASONING CHAINS

A key characteristic of a solution is its inherent fragility. A solution is not merely a collection of sentences; it is a delicate, logically-interlocked chain of reasoning where each step builds upon the previous one. A minor alteration to an early, correct step can invalidate the entire downstream logic. This fragility renders isotropic perturbation methods like `TARE` ineffective, as uniform paraphrasing would inevitably disrupt the "correct reasoning backbone," creating a noisy and uninformative loss landscape.

This very structure—a stable, correct reasoning backbone combined with a specific, identifiable flaw—makes the problem of solution optimization an ideal application for the `ATARE` framework. `ATARE` is fundamentally designed to handle textual components with varying degrees of sensitivity. The logical chain of a solution presents a natural, clear-cut case of anisotropic sensitivity, making `ATARE` not just a possible tool, but a perfectly suited one.

Our approach applies the core components of the `ATARE` lifecycle to this problem as follows:

**1. Semantic Sensitivity Analysis and Anisotropic Neighborhood Definition.**    The first step in the `ATARE` lifecycle is sensitivity analysis. We implement this by performing a comprehensive semantic diagnosis of the entire incorrect solution to identify the single, core "cognitive trap." This diagnosis effectively partitions the solution into two distinct regions of sensitivity:

- **Low-Sensitivity Region:** The identified core logical flaw itself. This part is considered the primary target for perturbation. The rationale is that a single type of logical error can manifest in many different, deceptive forms. All solutions that commit the same conceptual error, regardless of phrasing, are considered to be within the same **anisotropic semantic neighborhood**.

- **High-Sensitivity Region:** The entire chain of correct reasoning that precedes the flaw. This logical backbone is treated as immutable to maintain the solution's structural integrity.

**2. Inner Adversarial Search.** With the sensitivity regions defined, we conduct an inner adversarial search within this highly constrained neighborhood. To formalize this, let $s_t$ be the original incorrect solution at a given iteration $t$. We represent $s_t$ as a composition of two parts: its high-sensitivity correct backbone, denoted $s_{t,\text{correct}}$, and its low-sensitivity flaw, denoted $s_{t,\text{flaw}}$. The set of candidate solutions, $\mathcal{C}_{K_t}(s_t)$, is then generated by keeping the backbone fixed while using a generator to perturb only the flaw component. This process creates a set of $K_t$ candidates, defined as:

$$\mathcal{C}_{K_t}(s_t) := \left\{ s_{t,\text{correct}} \oplus \mathcal{G}^{(i)}(s_{t,\text{flaw}}) \right\}_{i=1}^{K_t}, \tag{20}$$

where the correct backbone $s_{t,\text{correct}}$ is held fixed, $\mathcal{G}_{\text{flaw}}$ is the generator responsible for creating variations of the flaw, the superscript $(i)$ indexes each of the $K_t$ unique generation events, and the symbol $\oplus$ denotes the composition of the text segments. This process explores the defined semantic neighborhood to find the variations that perform the worst on the task.

**3. Outer Robustness Update.** From the generated set of flaw variations, we identify the "worst-case" neighbor $s_{t,adv}^*$. The textual feedback derived from critiquing this worst-case scenario is then used to update the original solution $s_t$. By learning from the most challenging manifestation of its own core error, the solution is guided to patch this specific cognitive vulnerability. This directly implements the outer robustness update step of **ATARE**, optimizing for worst-case performance within the semantic neighborhood to guide the solution toward a flatter, more robust basin in the semantic landscape.

### D.2 EXPERIMENTS

**Tasks and Datasets.** We evaluate our solution optimization approach on two challenging benchmarks where model performance has not yet saturated.

- **GPQA** (Rein et al., 2023): The Google-proof Question Answering benchmark consists of expert-level multiple-choice questions in physics, biology, and chemistry. Its difficulty is highlighted by the performance gap between experts (81% accuracy) and skilled non-experts (22%).

- **MMLU** (Hendrycks et al., 2021): We use the challenging **College Physics** subset from the Massive Multitask Language Understanding benchmark, which is designed to measure human-level performance.

We follow the experimental setup of Revolve (Zhang et al., 2025) for iterative methods: we perform three iterations of optimization for each question and determine the final answer by majority voting. Consistent with prior work (Yuksekgonul et al., 2024), the evaluation metric is string-based exact match accuracy.

**LLM Backends and Counterparts.** We apply all methods on three distinct LLMs: **GPT-4o**, **Llama 3.1 8B Instruct**, and **Qwen 2.5 7B Instruct**. We compare our **ATARE**-based method against three primary baselines: **Zero-shot Chain-of-Thought (CoT)** (Kojima et al., 2023), **TextGrad** (Yuksekgonul et al., 2024), and **Revolve** (Zhang et al., 2025).

**Results and Analysis.** The performance of our method against the baselines is presented in Table 2. As shown, our **ATARE**-based approach consistently outperforms all baselines—CoT, TextGrad, and Revolve—across both datasets and all evaluated LLM backends. This strong and universal improvement validates our central hypothesis: for fragile, logically-interlocked reasoning chains, an anisotropic optimization strategy is superior. While first-order methods like TextGrad can sometimes struggle with the delicate structure of solutions, and even advanced methods like Revolve may not always escape local optima, our approach demonstrates a more robust path to improvement.

Table 2: **Results of solution optimization.** We report accuracy (%) and the relative improvement over Textgrad. The best and second-best results are highlighted with **bold** and underline, respectively.

| Dataset | Model | COT | TEXTGRAD | REVOLVE | ATARE |
|---|---|---|---|---|---|
| GPQA | GPT-4o | $48.5_{\downarrow 4.0}$ | $\underline{52.5}$ | $49.5_{\downarrow 3.0}$ | $\mathbf{53.0}_{\uparrow 0.5}$ |
| | Llama 3.1 8B Instruct | $27.8_{\downarrow 3.0}$ | $\underline{30.8}$ | $\underline{30.8}_{\uparrow 0.0}$ | $\mathbf{33.8}_{\uparrow 3.0}$ |
| | Qwen 2.5 7B Instruct | $35.9_{\downarrow 1.5}$ | $\underline{37.4}$ | $\underline{37.4}_{\uparrow 0.0}$ | $\mathbf{39.9}_{\uparrow 2.5}$ |
| MMLU (College Physics) | GPT-4o | $91.0_{\downarrow 2.5}$ | $93.5$ | $\underline{94.1}_{\uparrow 0.6}$ | $\mathbf{96.1}_{\uparrow 2.6}$ |
| | Llama 3.1 8B Instruct | $69.6_{\uparrow 0.0}$ | $69.6$ | $\underline{70.6}_{\uparrow 1.0}$ | $\mathbf{72.5}_{\uparrow 2.9}$ |
| | Qwen 2.5 7B Instruct | $\underline{78.4}_{\uparrow 0.0}$ | $78.4$ | $\underline{78.4}_{\uparrow 0.0}$ | $\mathbf{79.4}_{\uparrow 1.0}$ |

By precisely targeting only the low-sensitivity "flaw region" for perturbation while preserving the high-sensitivity correct reasoning, our approach provides a stable and effective optimization signal, consistently guiding the solution toward a more correct state.

## E  PROMPT DETAILS

This section provides the detailed architecture of the core prompts that power our optimization framework. We present the prompts for the main components of our framework—the perturbation generators for `TARE` and `ATARE`, and the LATO—as well as the prompts used for the Solution Optimization task.

### E.1  `TARE` PERTURBATION GENERATOR PROMPT

This prompt directs the `TARE` perturbation generator to conduct the isotropic neighborhood search required by our framework. It instructs a powerful LLM to create a set of minimally-altered, semantically-equivalent variations of a given text. The key to this process is the explicit goal of finding weaknesses; the prompt directs the generator to explore potentially worse-performing variations. This serves as the inner adversarial search, designed to identify sharp cliffs in the semantic landscape where the system's performance is brittle.

---

**`TARE` Perturbation Generator Prompt**

You are an expert in semantics and creative writing. Your task is to generate $\{k\}$ slightly different versions of the following text.
These perturbed versions must adhere to these rules:

1. **Maintain Core Intent:** The core intent and theme of the original text must be preserved.
2. **Small Degree of Perturbation:** The changes should be minor. For example, you can replace a few non-essential words, make small adjustments to sentence structure, or add/remove a few descriptive words.
3. **Preserve Factual Correctness:** Do not introduce irrelevant information or factual errors.
4. **Explore Vulnerabilities:** The goal is to explore closely related, but potentially worse-performing, variations of the text.

**Input Text**  ``system_prompt_text''

**Output Format**  The output should be a Python-style list of strings, with each string being one perturbed version.

Now, provide the $\{k\}$ perturbed versions for the original text.

---

### E.2  `ATARE` PERTURBATION PROMPT

The `ATARE` prompt architecture implements our framework's anisotropic search. It operates as a two-step "analyze-then-generate" chain. The first prompt performs Semantic Sensitivity Estimation,

directing an analyst LLM to decompose a prompt into three tiers of sensitivity (Constraint, Method, Style). The second prompt then performs Anisotropic Perturbation, using this analysis to guide a generator LLM in applying targeted, differentiated edits to each tier.

---

**Step 1: Sensitivity Analysis Prompt**

You are a specialized Prompt Architecture Analyst. Your task is to analyze a system prompt and decompose it into a three-tier hierarchy of components based on their sensitivity.

**Definition of Tiers**

- **Tier 1 (Constraint Layer - High Sensitivity):** Non-negotiable rules that define success or failure. Changing these will likely break the prompt's core function or format. (e.g., format rules, core task definition, absolute prohibitions).
- **Tier 2 (Method Layer - Medium Sensitivity):** Guidelines on "how" to perform the task. Changing these affects the quality and reasoning path, but not task completion itself. (e.g., "think step by step", process instructions).
- **Tier 3 (Style Layer - Low Sensitivity):** Persona, tone, and other stylistic elements. Changing these affects the prompt's personality, not its logic. (e.g., 'You are a helpful assistant', politeness).

**Input Prompt**   ``system_prompt_text''

**Output Format**   Your output MUST be a single, valid JSON object with three keys: ``constraint_layer'', ``method_layer'', and ``style_layer''. Each key must have a list of strings as its value.

Now, provide the three-tier JSON analysis for the original prompt.

---

**Step 2: `ATARE` Perturbation Generator Prompt**

You are an expert in semantics and creative writing. The goal is to explore closely related, but potentially worse-performing, variations of the text.

**Inputs**

1. **Original Prompt:** ``system_prompt_text''
2. **Three-Tier Sensitivity Analysis:** {analysis_text}

**YOUR TASK & RULES**   Your generated versions MUST adhere to these rules:

1. **Maintain Core Intent (Global Constraint):** All perturbed versions MUST maintain the core intent of the original prompt. The goal is to create semantically close variations to find weaknesses, not to write a new prompt.
2. **Targeted & Differentiated Perturbation:** Your changes should be targeted, and their degree must be based on the component's sensitivity from the analysis:
    - For **Tier 1 (Constraint Layer - High Sensitivity)** components, apply only MINIMAL and SUBTLE changes (e.g., synonym swaps like "only" to "just", slight rephrasing). These are fragile and require careful stress-testing.
    - For **Tier 2 (Method Layer - Medium Sensitivity)** components, you can apply MODERATE changes (e.g., rephrasing the reasoning process, altering the sequence of steps).
    - For **Tier 3 (Style Layer - Low Sensitivity)** components, you have the most freedom. Apply CREATIVE and DIVERSE changes (e.g., completely changing the persona, tone, or conversational style).
3. **No Invalid Information:** Do not introduce irrelevant information, contradictions, or factual errors.

Now, provide the {k} targeted, perturbed versions for the original text, strictly following all the rules above.

---

### E.3 LATO Optimizer Prompt

The LATO prompt is the engine for the Outer Robustness Update step. It is composed of three main parts: a glossary that defines the structured tags, a system prompt that outlines the optimizer's core task, and an instantiated user message that provides the specific context for an optimization step.

#### Glossary

To ensure the optimizer LLM correctly interprets the structured inputs, we first provide it with a glossary defining all the tags used in the prompt.

---
**LATO Prompt Glossary**

- `<ORIGINAL_VARIABLE>`: The original variable that you need to improve.
- `<PERTURBED_VARIABLE>`: A slightly perturbed version of the original variable that resulted in the feedback.
- `<ORIGINAL_VARIABLE_LOSS>`: The performance of the original variable on the current batch.
- `<PERTURBED_VARIABLE_LOSS>`: The performance of the perturbed variable on the current batch.
- `<LM_SYSTEM_PROMPT>`: The system prompt for the language model.
- `<LM_INPUT>`: The input to the language model.
- `<LM_OUTPUT>`: The output of the language model.
- `<FEEDBACK>`: The feedback to the variable.
- `<CONVERSATION>`: The conversation history.
- `<FOCUS>`: The focus of the optimization.
- `<ROLE>`: The role description of the variable.

---

#### LATO System Prompt

The LATO system prompt is designed to make the optimizer LLM explicitly landscape-aware. By providing a rich, contextual view of the local semantic landscape and the nature of a performance failure, it enables a more informed update than first-order methods, steering the variable towards a flatter, more robust semantic basin.

---
**LATO System Prompt**

You are an expert optimizer and a creative critic within an advanced AI system. You will be asked to creatively and critically improve text-based variables (prompts, solutions, code, etc.) to make them more effective and robust.

**THE PROCESS**    To do this, you will be given an `<ORIGINAL_VARIABLE>`. This variable was perturbed into a `<PERTURBED_VARIABLE>`, and the system's performance using this perturbed version resulted in critical `<FEEDBACK>`.

**YOUR TASK & OBJECTIVES**    Based on all available information, your goal is to generate a new, improved version of the `ORIGINAL_VARIABLE`. The new version must achieve the following objectives:

1. **Address the Failure:** It must resolve the specific issues pointed out in the provided `<FEEDBACK>`.
2. **Preserve Performance:** It must maintain or improve upon the original variable's good performance.
3. **Enhance Robustness:** It must be more resilient to similar small perturbations in the future.

**GUIDING PRINCIPLES**

---

- **Preserve Core Meaning:** Whatever the edit, you must strictly preserve the core task intent and local coherence of the original text.

- **Analyze Noisy Feedback:** The provided `<FEEDBACK>` may be noisy. Critically evaluate it to identify what is important and correct.

- **Consider Full Context:** Always pay attention to the variable's `<ROLE>` and the full context in which it is used to ensure your improvements are relevant.

**IMPORTANT - OUTPUT FORMAT** You MUST give your response by sending the improved variable between `{new_variable_start_tag}{improved variable}{new_variable_end_tag}` tags. The text you send between the tags will directly replace the variable. GLOSSARYTEXT

EXAMPLE OF AN INSTANTIATED LATO PROMPT

The following box shows an example of a complete prompt constructed and sent to the optimizer LLM, combining the system prompt with the specific variables for a single optimization step.

---

**Example of an Instantiated LATO Prompt**

**Here is the role of the variable you will improve:** `<ROLE>structured system prompt to a language model</ROLE>`.

- **The ORIGINAL variable that we are optimizing is:**

```
<ORIGINAL_VARIABLE>
You will answer a reasoning question. Think step by step. The
last line of your response should be of the following format:
``Answer: $VALUE'' where VALUE is a numerical value.
</ORIGINAL_VARIABLE>
<ORIGINAL_VARIABLE_LOSS> 1, 1, 1 </ORIGINAL_VARIABLE_LOSS>
```

- **When this variable was slightly perturbed into the following version:**

```
<PERTURBED_VARIABLE>
You are to solve a reasoning question. The final line of your
response should be in the format: ``Answer: $VALUE'' where
VALUE is a numerical value.
</PERTURBED_VARIABLE>
<PERTURBED_VARIABLE_LOSS> 1, 1, 1 </PERTURBED_VARIABLE_LOSS>
```

**The system received the following feedback based on the PERTURBED version's performance:**

```
<CONTEXT>
Here is a conversation:
<CONVERSATION>
...
<LM_INPUT>
I have three oranges, a pig, a frog, a cow, three bananas,
a nectarine, and a snail. How many animals do I have?
</LM_INPUT>
<LM_OUTPUT>
To find the total number of animals, we need to identify the
animals...So, the total number of animals is 4.
Answer: 4
</LM_OUTPUT>
</CONVERSATION>
Here is the feedback we got in the conversation:
<FEEDBACK>
```

```
To improve the adaptively perturbed prompt variable in order
to enhance the objective function, consider the following
strategies:
...
</FEEDBACK>
</CONTEXT>
```

**Based on the feedback from the perturbed version, improve the ORIGINAL variable to make it more robust.**

### E.4  SOLUTION OPTIMIZATION PROMPT DETAILS

This section details the prompt architecture for our `ATARE`-based solution optimization pipeline. The process is divided into two main stages, each with a corresponding prompt.

The first stage is Flaw Diagnosis, which operationalizes the Semantic Sensitivity Analysis step. The prompt below instructs an expert LLM to identify the core "cognitive trap" in an incorrect solution, thereby defining the low- and high-sensitivity regions.

---

**Flaw Diagnosis Prompt**

You are a world-class expert in logic and science. The following solution is INCORRECT. Your task is to deeply analyze its reasoning and clearly describe the core "cognitive trap" or "flawed reasoning path" that led to the wrong conclusion.

**Incorrect Solution to Diagnose ($s_t$)**  `{solution_var.value}`

Please provide your detailed analysis of its flawed reasoning path.

---

The second stage is the Anisotropic Adversarial Search, executed by the `ATARE` Perturbation Generator. This prompt takes the flaw analysis from Stage 1 as input and directs the LLM to generate diverse variations of only the identified flaw, while strictly preserving the correct reasoning backbone.

---

**`ATARE` Perturbation Generator Prompt**

You are a creative AI that can mimic different thinking styles. You will receive an incorrect solution and an analysis of its core flaw. Your task is to generate {k} new, distinct, but equally flawed solutions.

**Guiding Principle**  Treat the solution as a reasoning chain. The correct reasoning *before* the identified flaw is the high-sensitivity backbone that **MUST** be preserved. Your task is to vary the expression of the flaw itself (the low-sensitivity target).

**Rules**  All new solutions **MUST**:

1. Commit the same type of core error as described in the "Flaw Analysis".

2. Use different phrasing, examples, or intermediate steps to express this error. The goal is to explore different deceptive manifestations of this single cognitive trap.

3. Ensure that the final conclusion and the chosen answer letter **LOGICALLY FOLLOW** from your flawed reasoning.

4. Choose your final answer from the available options in the "Problem Context". **Do not** invent new options.

**Inputs**

• **Problem Context:** {question}

• **Original Incorrect Solution ($s_t$):** {solution_var.value}

• **Flaw Analysis Report:** {flaw_analysis}

**Output Format**  The output **MUST** be a valid Python list of strings.

---

# F NOTATIONS

We present a comprehensive review of the commonly used notations and their definitions in Tab. 3.

| Notation | Definition |
|---|---|
| $\mathcal{M}$ | The black-box LLM. |
| $\mathcal{P}$ | The discrete semantic space of textual prompts. |
| $\mathcal{D}$ | The dataset. |
| $p$ | A textual prompt. |
| $\ell(\cdot, \cdot)$ | The loss function (or evaluator $\mathcal{E}$) mapping outputs to numeric scores. |
| $\mathcal{L}_{\mathcal{D}}(p)$ | The empirical prompt risk on dataset $\mathcal{D}$. |
| $d_{\text{text}}(p, p')$ | The semantic dissimilarity between prompts $p$ and $p'$. |
| $\rho_{\text{text}}$ (or $\rho_t$) | The radius of the semantic neighborhood. |
| $B(p, \rho_{\text{text}})$ | The isotropic semantic neighborhood of prompt $p$. |
| $\mathcal{L}_{\text{S}}(p, \rho_{\text{text}})$ | The textual sharpness-aware loss. |
| $\mathbf{W}_p$ | The diagonal weight matrix for anisotropic sensitivity. |
| $d_{\text{ani}, \mathbf{W}_p}(p, p')$ | The anisotropic semantic distance metric. |
| $B_p(p, \rho_{\text{text}})$ | The anisotropic ellipsoidal neighborhood. |
| $s_{t,j}$ | The sensitivity score of the $j$-th prompt component. |
| $\beta$ | The exponent parameter controlling sampling probability in ATARE. |
| $\mathcal{G}$ | The generator oracle used for sampling paraphrases. |
| $\mathcal{O}$ | The optimizer oracle used for proposing updates. |
| $t$ | The current iteration index. |
| $\mathcal{B}_t$ | The minibatch of data at iteration $t$. |
| $K_t$ | The budget (number of samples) for the inner adversarial search. |
| $\mathcal{C}_{K_t}(p_t)$ | The set of candidate prompts sampled in the inner loop. |
| $p_{t,\text{adv}}^{\star}$ | The adversarial (worst-case) prompt neighbor. |
| $\widehat{\mathcal{L}}$ | The estimated empirical loss on a specific batch. |
| $M_t$ | The budget (pool size) for the optimizer proposals. |
| $\delta_t$ | The semantic budget constraint for the optimizer. |
| $\mathcal{U}_{M_t}(p_t)$ | The improvement pool generated by the optimizer. |
| $\tilde{K}$ | The sampling budget for the outer robust validation. |
| $\eta$ | The tolerance threshold for accepting an update. |
| $\Pi_{\text{LATO}}$ | The meta-prompt template for the Landscape-Aware Textual Optimizer. |

Table 3: Notation and Definitions

