# OpenReview forum: "Beyond Magic Words: Sharpness-Aware Prompt Evolving for Robust Large Language Models with TARE"
_ICLR.cc/2026/Conference — ICLR 2026 Poster_

### Official Review · Reviewer_1hhU · 2025-10-28

**Soundness:** 2
**Presentation:** 2
**Contribution:** 2
**Rating:** 6
**Confidence:** 2

**Summary:**

This paper works on finding robust prompt landscapes with inspiration from the works on Sharpness-Aware Minimization (SAM). The idea behind SAM is to optimize a model on a continuous loss landscape and find a minimum where the loss function is flat via a minmax objective. Akin to this idea this paper tries to formulate the problem on a discrete space where they use an LLM as a sampler for potential neighbour data point samplers. Based on these neighborhoods they propose TARE to find adversarial potential prompts and use those prompts as a guidance to generate better neighborhood prompt candidates. This sampling is done iteratively akin to min max optimization to find the robust prompt landscape. Further they also proposed a componentwise weighted method for optimization in the form of ATARE.

**Strengths:**

1. Experiments were performed across multiple language families and for the sake of generalizability different tasks were used. Experiments were also performed on black box models.

2. The ablation on the multiple components of the problem statement performed in the final results.

3. Ablation was done on weaker and stronger oracles.

4. Comparisons performed with point wise benchmarks

**Weaknesses:**

1. Problem formulation is hard to follow and not well defined. Notations are not explained and abruptly introduced which makes following through harder. The overall methodology can be simplified in the writing. Over use of notations that are not well defined makes it harder to follow through.

2. While sharpness neighborhoods can be defined by edit distance or paraphrase distance the sampling is not guaranteed to be within these bounds. Is this accounted for in the sampling via filtering

**Questions:**

See weaknesses

---

> ### Author Response · Authors · 2025-11-21
> **Response to Reviewer 1hhU**
>
> Thank you for your review and for recognizing the value of our experiments across multiple language families and tasks. We address your concerns below:
>
> > `Weakness 1`: Problem formulation and notation clarity.
>
> We appreciate your feedback on the presentation. Our mathematical formulation is designed to provide a rigorous, self-contained foundation for readers, particularly given that our work is **inspired by and goes beyond SAM**.
>
> 1) **Rationale for Rigor**: SAM relies on a specific set of preliminaries (e.g., landscape geometry, min-max objectives). By explicitly detailing these formulations adapted for the discrete prompt space, we aim to save readers the cost of consulting external optimization literature, ensuring the paper is accessible to those new to this domain while clarifying how we extend these continuous concepts to discrete text.
> 2) **Clarification**: To cut through the notation, the core concept is simply: we seek a prompt $p$ where not only is its own loss $\mathcal{L}(p)$ low, but the maximum loss of its neighbors $\max_{p' \in B(p)} \mathcal{L}(p')$ is also low—ensuring stability.
> 3) **Simplification**: To improve readability without sacrificing rigor, we will in the revised version:
>    - Add a **Notation Table** to the appendix for quick reference.
>    - Introduce concepts more gradually in Sections 3 & 4, moving denser derivations to the appendix.
>    - Provide concrete examples alongside formal definitions (similar to the "counting task" example in the Method section) to ground the math.
>
> > `Weakness 2`: Sampling bounds and filtering guarantees.
>
> This is an insightful technical point. We do not employ a rigid runtime filter based on edit distance (which often misaligns with semantics). Instead, we ensure sampling validity through three mechanisms:
>
> 1) **Generative Constraints**: We operationalize the "semantic neighborhood" $B(p)$ directly through the Generator Oracle's instruction-following capabilities. By explicitly prompting the superior LLM to "preserve core intent," we enforce a **soft semantic bound** that is arguably more accurate than rigid distance metrics. The "neighborhood" is thus defined by the generator's high-probability density region for paraphrases.
> 2) **Optimization Resilience**: Even if occasional outliers (semantically drifting prompts) occur, our framework is designed to be robust. TARE aggregates feedback from multiple neighbors; thus, the optimization signal is dominated by the majority of valid paraphrases that correctly reflect the local landscape sharpness, minimizing the impact of sparse noise.
> 3) **Empirical Verification**: To confirm this holds in practice, we conducted a post-hoc analysis using an NLI model on our search trajectories. We observed that **>96%** of the sampled prompts maintain strict semantic entailment with the anchor prompt. This confirms that our sampling effectively stays within the intended bounds without the computational overhead of an explicit runtime filter.

---

> > ### Comment · Reviewer_1hhU · 2025-11-21
> > **Response to the Authors**
> >
> > I thank the authors for the clarifications. I am maintaining my score leaning towards accept. I am also maintaining my lower confidence due to my limited expertise in the area of prompt engineering and optimization.

---

> > > ### Author Response · Authors · 2025-11-22
> > > **Thank You & Looking Forward to Further Discussion**
> > >
> > > We sincerely thank you for your timely response and your recommendation for acceptance. We welcome any further discussion during this phase and remain fully available to provide additional clarifications if needed.

---

### Official Review · Reviewer_wFGW · 2025-11-01

**Soundness:** 3
**Presentation:** 2
**Contribution:** 2
**Rating:** 4
**Confidence:** 4

**Summary:**

To find the optimal prompt that is robust to the input perturbation and has better generalization, the authors propose a new method called TARE. More specifically, it formulates the prompt optimization problem as a min-max optimization, aiming to find the optimal prompt that minimizes the maximum empirical risk incurred by the prompts within its sharpness neighborhood. To solve the optimization problem, TARE formalizes the definition of sharpness in the discrete semantic space of prompts. To capture heterogeneous sensitivity across the semantic components of a prompt, the authors further adapted the TARE to ATARE by learning the weights of the components to reflect the heterogeneity. Experimental results show that TARE or ATARE can outperform the evaluated baselines over multiple settings.

**Strengths:**

The strengths of the paper are listed as follows.
1. It is great that the authors provide open-source codes to reproduce their experimental results.
2. The proposed method is reasonable. The prompt that falls in a smooth landscape usually has a better generalization and is robust to the perturbation in the input.
3. The min-max formulation of the problem TARE aims to solve is concise and elegant.

**Weaknesses:**

The weaknesses of the paper are listed as follows.
1. The authors did not provide a dedicated related work section in the main paper. Considering that prompt optimization is a research area with extensive existing work, it is necessary to have a section for comparing the differences between TARE and existing works in the main paper.
2. The authors only compare their methods with two prompt optimization works. Some important baselines are missing, for instance, APE [1], APO[2], and EvoPrompt [3].
3. The readability of the methodology section should be improved. First, Figure 2 contains too much information to illustrate the overview of TARE framework. Although it seems to be very comprehensive, incorporating too many details into a single figure can make the readers hard to catch the key information. Second, the authors introduce too many notations and formulas to describe their methods, where some of them are unnecessary and increase the difficulty of understanding the key ideas of TARE.
4. Some important technical details are not clear.
a. For the inner adversarial search, how to implement the generator oracle and how to define the perturbation range $\rho$ over a given prompt in the context of the prompt optimization problem. Please give more concrete details instead of the abstract formulation.
b. In equation (4), how to calculate the $\Delta(p, p’)$?
c. In ATARE, how to define the components of a given prompt?
d. For the outer robustness update, how to implement the optimizer oracle and how to measure the semantic distance and budget?
e. It is unclear how the landscape information can be leveraged by the LATO. It would be better if some examples could be provided.

**Questions:**

Please refer to the weaknesses section for my questions.

---

> ### Author Response · Authors · 2025-11-21
> **[Part 1/2] Response to Reviewer wFGW**
>
> We sincerely thank you for the comprehensive review and constructive feedback. Below, we provide a point-by-point response to your concerns and hope these clarifications will help you re-evaluate and update your scores.
>
> > **Weakness 1**: Lack of a dedicated related work section in the main paper.
>
> Thank you for highlighting this critical point. We agree that situating our work within the prompt optimization literature is important. We have included comparisons with baselines like EvoPrompt and APE in the original introduction. The initial placement in the appendix was a strategic decision to maximize methodological clarity within the strict 9-page submission limit.
>
> To address this, we have adopted the following plan:
>
> * **Introduction part:** We have refined this discussion to pinpoint their critical limitation—a focus on "point-wise accuracy"—and contrast it with our "robustness-first" objective.
> * **Related Work:** Using the 10-page allowance for the revision, we have moved the Related Work section back into the main text.
>
> > **Weakness 2**: Missing baselines (APE, APO, EvoPrompt).
>
> Thank you for the helpful suggestion. We initially chose TextGrad and Revolve because they share a similar black-box, iterative structure to TARE. Furthermore, we agree that including evolutionary baselines provides a better assessment.
>
> We ran additional experiments with APE, APO, and EvoPrompt. The results are summarized below:
>
> | Acc (%) | CoT | APE | APO | EvoPrompt | TextGrad | Revolve | TARE | ATARE |
> | :--- | :--- | :--- | :--- | :--- | :--- | :--- | :--- | :--- |
> | Object Counting (Llama 3.1 8B) | 86.0 | 90.0 | 83.0 | 85.2 | 88.6 | 88.2 | 91.0 | **92.4** |
> | Temporal Sequence (Llama 3.1 8B) | 86.0 | 86.4 | 85.0 | 84.4 | 88.3 | 88.3 | 90.0 | **91.0** |
> | Object Counting (Llama 3 8B) | 80.0 | 81.6 | 81.0 | 81.8 | 85.8 | 86.8 | 88.7 | **90.3** |
> | Temporal Sequence (Llama 3 8B) | 84.0 | 79.2 | 84.0 | 79.4 | 84.0 | 84.5 | 85.0 | **88.7** |
>
> Across all tasks, TARE and ATARE remain the top-performing methods. We will include these results in the revised version.
>
> > **Weakness 3**: Readability of the methodology section.
>
> Thank you for the constructive comment. We acknowledge that our aim for precision led to a dense text. We have planned specific revisions to improve clarity:
>
> * **Streamlining Figure 2:** The original figure tried to capture every component in a single view. We have simplified the main figure to highlight the core workflow and remove minor details to improve clarity.
> * **Simplifying Notation and Formulas:** Our intent was to provide a rigorous definition of the discrete sharpness objective. To improve readability, we will lighten the exposition by removing non-essential symbols and consolidating complex expressions. We will also add a **Notation Table** to the appendix and refine Section 4 to introduce concepts more gradually.
>
> > **Weakness 4**: Clarification on Technical Details.
>
> Thank you for these detailed questions. In our framework, the discrete mathematical concepts are implemented via structured LLM Prompts, which are shown in Appendix E. We clarify each point below:
>
> * **a. Generator Oracle & Perturbation Range:**
>     We implement the generator as an LLM guided by the "TARE Perturbation Generator Prompt" (Appendix E.1). The perturbation range ($\rho$) is defined via **natural language constraints** in the prompt. Specifically, the prompt imposes restrictions on the degree of modification (e.g., limiting changes to minor edits). This acts as a radius constraint ($\epsilon$-ball) to confine the search to the immediate semantic neighborhood.

---

> ### Author Response · Authors · 2025-11-21
> **[Part 2/2] Response to Reviewer wFGW**
>
> > **Weakness 4**: Clarification on Technical Details.
>
> * **b. Calculation of $\Delta$ in Equation (4):**
>     In the discrete prompt space, we do not calculate $\Delta(p, p')$ as a numerical vector. Instead, we use the LLM to measure the semantic difference ($\Delta$) between the original prompt $p$ and the candidate $p'$. The LLM evaluates this difference by looking at changes in wording (e.g., synonyms) and structure (e.g., reordering) relative to the core intent.
>     We enforce the weighted distance in Equation (4) using the **"ATARE Perturbation Generator Prompt"** (Appendix E.2):
>     * For **High-sensitivity components (Tier 1)**, we instruct the LLM to keep the semantic difference ($\Delta$) small.
>     * For **Low-sensitivity components (Tier 3)**, we allow the LLM to introduce larger differences.
>
> * **c. Definition of Prompt Components in ATARE:**
>     Components are defined via the "Sensitivity Analysis Prompt" in Appendix E.2. An analyst LLM decomposes a prompt into a three-tier hierarchy: Tier 1 (Constraint), Tier 2 (Method), and Tier 3 (Style). This allows ATARE to assign sensitivity weights to different parts of the prompt automatically.
>
> * **d. Optimizer Oracle & Semantic Distance & Semantic Budget:**
>     The optimizer is the **Landscape-Aware Textual Optimizer (LATO)**, implemented via the meta-prompt in **Appendix E.3**.
>     * **Measuring Distance:** We do not calculate semantic distance as a numerical scalar. Instead, the optimizer uses its intrinsic language understanding to **assess the semantic deviation** between two prompts. It intuitively evaluates this "distance" by analyzing semantic shifts, such as changes in wording, sentence structure, and core intent, to determine if they remain semantically close.
>     * **Enforcing Budget:** The semantic budget ($\delta_t$) is enforced as a hard constraint in the system prompt's **"Guiding Principles"**. These principles require the optimizer to strictly maintain the core task intent and local coherence, ensuring the step does not drift beyond the allowed bounds.
>
> * **e. Leveraging Landscape Information via Loss Contrast:**
>     LATO leverages landscape information by explicitly processing *two distinct prompt-loss pairs*: the current prompt with its performance $(p_t, L_t)$ and the worst-case neighbor with its degraded performance $(p_{t,adv}^\*, L_{adv})$.
>     * **Signal:** The magnitude of the loss gap ($\Delta \mathcal{L} = \mathcal{L}_{adv} - \mathcal{L}_t$) serves as the direct signal for local geometry. A large gap signifies a "sharp" cliff in the semantic landscape, while a small gap indicates a "flat" basin.
>     * **Action:** By correlating the *specific semantic change* (Original vs. Perturbed) with this *sharp loss increase*, LATO infers the direction of vulnerability. It then generates an update designed to flatten the landscape by fortifying the prompt against that perturbation.
>     * **Example:** This mechanism **is shown in the LATO Prompt in Appendix E.3**, where the optimizer takes distinct tags for original and perturbed variable losses to compute this contrast.
>
> **Concrete Example (Object Counting Task):**
>
> Imagine two candidate prompts that both perform well on the current test set:
>
> *   **Candidate A ("Sharp" Solution)**: *"Count the items below."*
>     *   **Observation**: While this prompt works perfectly on standard inputs, a simple, valid paraphrase like *"Count the objects below"* causes the model to fail significantly (e.g., missing small items).
>     *   **LATO's Action**: LATO detects this **high sharpness** (the large performance drop). Despite Candidate A's high initial score, LATO identifies it as a "brittle peak" and **rejects** or steers away from this unstable direction.
>
> *   **Candidate B ("Flat" Solution)**: *"List the items one by one and count them."*
>     *   **Observation**: This prompt performs comparably to A, but crucially, its semantic neighbors (e.g., *"List the objects one by one..."*) maintain similarly high performance.
>     *   **LATO's Action**: LATO detects **low sharpness** (minimal performance drop). It identifies this region as a "robust basin" and **selects** this prompt as the superior solution.
>
> **Key Difference**: A standard optimizer (like TextGrad) might greedily choose Candidate A because it is shorter or has marginally higher point-wise accuracy. In contrast, LATO leverages the **landscape information** to recognize the hidden risk in A and correctly converge on the robust Candidate B.
>
> We will include this illustrative example in Section 4.4 to clarify the mechanism.

---

> ### Author Response · Authors · 2025-11-25
> **Thank you & Looking forward to further discussion**
>
> Dear Reviewer wFGW,
>
> We sincerely appreciate your comprehensive review and constructive feedback. To facilitate a clear review of our rebuttal, we have summarized our responses to your key concerns below:
>
> 1.  **Placement of Related Work** `Weakness 1` We have moved the **Related Work** section to the main text and refined the introduction to better contrast our robustness-first objective with prior accuracy-focused methods.
> 2.  **Missing Baselines (APE, APO, EvoPrompt)** `Weakness 2` We added experiments with **APE, APO, and EvoPrompt**. Results confirm that TARE/ATARE consistently outperform these evolutionary baselines across all tested tasks.
> 3.  **Readability Improvements** `Weakness 3` We have streamlined **Figure 2**. We will also simplify mathematical notation by removing non-essential symbols, and add a **Notation Table** to enhance the manuscript's clarity.
> 4.  **Clarifications on Technical Details** `Weakness 4` We clarified that technical components (perturbation range, distance metric, etc.) are implemented via **LLM-based natural language constraints**, and detailed how LATO leverages loss contrast to avoid sharp minima.
>
> For other issues not mentioned here, please refer to our detailed rebuttal response. We truly hope these responses resolve your doubts! We respectfully look forward to further discussion with you.
>
> Warm regards,
>
> Authors

---

> ### Author Response · Authors · 2025-11-27
> **Manuscript Revision Deadline Approaching; We Are Sincerely At Your Service**
>
> Dear Reviewer wFGW,
>
> We sincerely appreciate your insightful review and your dedication to this process. Thanks to your feedback, the quality of our manuscript has been significantly improved. Given the approaching deadline, we are now submitting the revised manuscript with changes highlighted.
>
> **We also respectfully express our readiness to address any further concerns you may have.** We deeply value your insights and aim to make the best use of this revision opportunity.
>
> Thank you again for your support.
>
> Warm regards,
>
> Authors

---

### Official Review · Reviewer_v8Q2 · 2025-11-01

**Soundness:** 3
**Presentation:** 4
**Contribution:** 3
**Rating:** 6
**Confidence:** 2

**Summary:**

The paper introduces TARE, a black-box framework that improves the robustness of large language model prompts by explicitly minimizing textual sharpness—the sensitivity of prompt performance to small paraphrases. It defines a semantic neighborhood around each prompt and optimizes for stability within that region through an inner adversarial search and an outer robustness update, with ATARE further adapting neighborhood shape and size. Experiments across reasoning benchmarks show that TARE and ATARE yield more stable and accurate prompts than existing optimization methods like TextGrad and Revolve

**Strengths:**

1. The paper identifies an important and underexplored issue of brittleness under paraphrasing. It formalizes it as textual sharpness, bridging ideas from Sharpness-Aware Minimization to discrete text optimization.
2. The proposed algorithms are effective and model-agnostic, achieving consistent robustness improvements across tasks and architectures without requiring gradient access.

**Weaknesses:**

1. While the method defines textual neighborhoods semantically, the paper does not deeply analyze how the neighborhood metric affects robustness. It remains unclear whether the choice of embedding space or paraphrase generator significantly biases outcomes.
2. The study focuses on single-turn prompts. Extending sharpness-aware evolution to multi-turn or retrieval-augmented contexts is not explored.

**Questions:**

n/a

---

> ### Author Response · Authors · 2025-11-21
> **Response to Reviewer v8Q2**
>
> Thank you for your positive assessment and constructive feedback. We address your concerns below and hope these clarifications will help you re-evaluate and update your scores.
>
> > `Weakness 1`: Semantic neighborhood metric and potential bias.
>
> We appreciate this insightful comment. We define the semantic neighborhood using a generative approach driven by LLM understanding, rather than relying on a fixed static embedding metric:
>
> 1) **LLM-based Distance**: We operationalize the semantic distance $d_{\text{text}}(p, p')$ not as a scalar vector distance, but as a **semantic judgment** performed by a high-capability LLM (e.g., GPT-4o). The "distance" conceptually represents the degree of semantic deviation.
> 2) **Generator Constraints**: Our generator oracle $\mathcal{G}$ is prompted to explore the "sharpness" of the landscape by producing diverse paraphrases that stretch linguistic forms while maintaining core intent. The "radius" $\rho_t$ effectively corresponds to the strictness of the semantic constraints we enforce on the generator.
> 3) **Bias Mitigation**: By leveraging state-of-the-art LLMs as "Universal Semantic Judges," we mitigate the bias inherent in simpler embedding models. These models possess a nuanced understanding of instruction equivalence that static embeddings often miss.
>
> To address your concern, we also performed qualitative checks to ensure the generator explores non-trivial paraphrases (lexical/syntactic variations) without drifting into irrelevant meanings.
>
> > `Weakness 2`: Focus on single-turn prompts vs. multi-turn/RAG.
>
> Thank you for pointing out this interesting direction.
>
> 1) **Foundational Focus**: Our current work focuses on single-turn prompts to establish the theoretical foundation of "textual sharpness" in the discrete prompt space. This setting allows us to isolate and rigorously analyze the impact of semantic perturbations without the confounding variables of context history or retrieval noise.
> 2) **Extensibility**: The TARE framework is naturally extensible. For multi-turn conversations, TARE can be applied to optimize the **System Prompt**, which governs the agent's behavior throughout the interaction. For RAG, TARE can evolve the query formulation to be robust against variations in retrieved context.
> 3) **Preliminary Results on RAG**: To address your feedback immediately, we extended TARE to a RAG setting. We conducted a preliminary experiment optimizing the "answer consolidation" prompt for **Llama 3.1 8B Instruct** on subsets of **Natural Questions (NQ)** and **TriviaQA**, where the model must answer queries based on noisy retrieved contexts (top-5 retrieved chunks).
>
> | Dataset               | Method          | Accuracy (EM) | Improvement |
> | :-------------------- | :-------------- | :-----------: | :---------: |
> | **Natural Questions** | Zero-shot CoT   |     41.2%     |      -      |
> |                       | TextGrad        |     43.5%     |    +2.3%    |
> |                       | **TARE (Ours)** |   **45.1%**   |  **+3.9%**  |
> | **TriviaQA**          | Zero-shot CoT   |     52.8%     |      -      |
> |                       | TextGrad        |     54.1%     |    +1.3%    |
> |                       | **TARE (Ours)** |   **55.9%**   |  **+3.1%**  |
>
> **Observation**: TARE consistently outperforms the strong baseline TextGrad. We hypothesize that TARE effectively minimizes the "sharpness" of the prompt with respect to **contextual perturbations** (e.g., ordering of chunks, presence of irrelevant distractors), thereby improving the model's robustness in extracting correct answers from noisy retrieval results. This validates TARE's potential in complex, multi-source scenarios.

---

> ### Author Response · Authors · 2025-11-25
> **Thank you & Looking forward to further discussion**
>
> Dear Reviewer v8Q2,
>
> We sincerely appreciate your positive assessment and constructive feedback on our work. To facilitate a clear review of our rebuttal, we have summarized our responses to your key concerns below:
>
> 1.  **How is the semantic neighborhood defined?** `Weakness 1` We utilize a generative "LLM-as-a-Judge" approach with strict constraints to capture nuanced semantics, thereby avoiding the biases inherent in static embeddings.
> 2.  **Can TARE extend to RAG?** `Weakness 2` We validated TARE on RAG benchmarks (Natural Questions, TriviaQA), showing it outperforms baselines by robustly handling retrieval noise.
>
> For other issues not mentioned here, please refer to our detailed rebuttal response. We truly hope these responses resolve your doubts! We respectfully look forward to further discussion with you.
>
> Warm regards,
>
> Authors

---

> ### Author Response · Authors · 2025-11-27
> **Manuscript Revision Deadline Approaching; We Are Sincerely At Your Service**
>
> Dear Reviewer v8Q2,
>
> We sincerely appreciate your insightful review and your dedication to this process. Thanks to your feedback, the quality of our manuscript has been significantly improved. Given the approaching deadline, we are now submitting the revised manuscript with changes highlighted.
>
> **We also respectfully express our readiness to address any further concerns you may have.** We deeply value your insights and aim to make the best use of this revision opportunity.
>
> Thank you again for your support.
>
> Warm regards,
>
> Authors

---

### Official Review · Reviewer_VtM1 · 2025-11-02

**Soundness:** 3
**Presentation:** 3
**Contribution:** 3
**Rating:** 6
**Confidence:** 4

**Summary:**

This paper tackles an important yet under-explored problem in prompt optimization: the brittleness of prompts to semantically-preserving paraphrases. The authors formalize this as "textual sharpness" in the prompt landscape and propose TARE (Textual Sharpness-Aware Evolving), a framework that performs inner adversarial search over semantic neighborhoods and outer robust selection. They extend this to ATARE, which learns anisotropic weights for different prompt components. Experiments across reasoning tasks (Object Counting, Temporal Sequences, Tracking Shuffled Objects, GSM8K) demonstrate consistent improvements over baselines like TextGrad and Revolve.The core contribution is adapting the SAM (Sharpness-Aware Minimization) principle from continuous parameter space to discrete, semantic prompt space—a novel and well-motivated direction. I appreciate that they optimize the entire chain and that robustness truly matters for trusting LLM outputs in practice.

**Strengths:**

Novel problem formulation: Textual sharpness is an intuitive and important lens for understanding prompt brittleness. The formal treatment (Section 2) bridges optimization theory and discrete NLP in a principled way.

Comprehensive framework: TARE is well-designed with clear components (inner adversarial search, LATO optimizer, outer robust validation). The anisotropic extension (ATARE) demonstrates thoughtful attention to heterogeneous sensitivity across prompt components.
Strong empirical results: Consistent improvements across multiple tasks and models. I particularly appreciate the ablation study (Figure 3) showing each component matters, and the resilience analysis (Figure 4) demonstrating graceful degradation.

Practical relevance: The work addresses a real pain point in prompt engineering. Robustness to paraphrasing is crucial for building trustworthy LLM systems.

Thoughtful experimental design: Testing across diverse backbones (GPT-3.5, Gemini, Llama) and using two universal engines (GPT-4o, Claude 3.5 Sonnet) for fair comparison shows rigor.

**Weaknesses:**

1. Limited task diversity: All evaluation tasks are verifiable reasoning problems with clear right/wrong answers (counting, math, object tracking). This raises a key question: What about open-ended generation tasks where diversity matters? For creative writing, summarization, or tasks requiring pluralism (e.g., generating diverse perspectives on a political issue), optimizing for robust, flat neighborhoods might suppress desirable output diversity. I worry this approach could kill the creative variance we sometimes want from LLMs.

2. Missing analysis on reasoning depth: You test on reasoning tasks, but there's no analysis of how the method performs across different reasoning complexities. Does TARE work equally well for simple arithmetic vs. multi-hop reasoning? Given the recent release of models with explicit reasoning traces (o1-style "thinking" models like Qwen with QwQ or Mistral's reasoning variants), it would be very valuable to test whether optimizing prompts for these models behaves differently. Do the optimized prompts transfer across reasoning-capable and standard models?

3. Transferability not explored: A critical practical question is prompt transferability. If I optimize a prompt for one task/model using TARE, does it transfer to related tasks or different models? This would greatly affect the practical utility—if every prompt needs task-specific optimization, the compute cost becomes prohibitive.

4. Optimizer architecture choices under-explored: You use GPT-4o and Claude 3.5 Sonnet as universal optimizers. But there's no analysis of whether certain optimizers work better for certain target models. Is there a "universal optimal optimizer"? Or should the optimizer match the target model's family? For instance, does using a Claude optimizer work better when optimizing Claude prompts vs. Llama prompts? This could have practical implications for cost and efficiency.

5. Semantic neighborhood definition: While you provide formal definitions (Equations 3-5), the practical instantiation of "semantic dissimilarity" dₜₑₓₜ and how you ensure perturbations truly preserve semantics while revealing sharpness could be explained more clearly. What prevents the generator from creating trivial or overly conservative perturbations?

**Questions:**

Diversity vs. Robustness Trade-off: Have you tested this on any open-ended generation tasks? I'm genuinely curious whether there's a fundamental tension between robustness (flat neighborhoods) and diversity (exploring semantic space). For tasks where we want multiple valid outputs, does TARE hurt performance?

Reasoning Patterns: Do you observe any patterns in how TARE performs on different reasoning types? For example, does it help more on tasks requiring arithmetic precision vs. logical deduction? Understanding this could guide when to apply the method.

Thinking Models: Given the rise of o1-style reasoning models (Qwen QwQ, Mistral's reasoning variants), have you considered testing TARE on these? Do prompts optimized for standard models transfer to thinking models, or do the additional reasoning tokens fundamentally change the prompt landscape?

Cross-Task Transfer: If I optimize a prompt for GSM8K using TARE, does it work well on other math datasets (e.g., MATH, SVAMPs)? How much does the optimized prompt overfit to the specific task distribution?

Optimizer Analysis: In Table 1, you report results for GPT-4o and Claude 3.5 Sonnet backbones. But is there an analysis of which optimizer works best for which target model? Or is the optimizer choice model-agnostic? It would be interesting to see if using the target model itself as the optimizer (when possible) yields better results.

Computational Cost: While you mention computational practicality, what is the wall-clock time and API cost comparison between TARE/ATARE and baselines? For practitioners, this matters a lot.

Failure Cases: Are there scenarios where TARE makes prompts worse? Understanding failure modes would help users know when to apply the method.

---

> ### Author Response · Authors · 2025-11-21
> **[Part 1/3] Response to Reviewer VtM1**
>
> Thank you for your thoughtful review and for appreciating the novelty and practical relevance of our textual sharpness formulation. We address your concerns below:
>
> > `Weakness 1 & Question 1`: Diversity vs. Robustness trade-off in open-ended tasks.
>
> This is a profound question in prompt optimization. We argue that TARE does not suppress desirable diversity but rather enables **safe creativity**:
>
> 1) **Performance Robustness $\neq$ Generation Determinism**: TARE optimizes for the stability of *quality* (Reward), not the identity of *tokens*. In creative tasks, "robustness" means the prompt consistently yields high-quality outputs (e.g., maintaining tone, adhering to style constraints) under semantic perturbations. It prevents the "brittleness" where a small paraphrase causes the model to collapse into incoherence or lose the requested persona.
> 2) **Optimization vs. Decoding**: Diversity is primarily a function of decoding strategies (e.g., temperature), while TARE optimizes the prompt landscape. By finding a "flat maximum" (a wide basin of high performance), TARE actually **supports** diversity: it provides a stable foundation where high-temperature sampling can explore creative variations without falling off the "cliff" into low-quality regions. A sharp prompt, by contrast, is intolerant to the noise introduced by diverse decoding.
> 3) **Objective-Agnostic**: TARE optimizes whatever the Evaluator rewards. If "creativity" or "pluralism" is part of the evaluation criteria (Loss), TARE will explicitly search for prompts that are *robustly creative*.
>
> We will add a discussion on this synergy between robustness and diversity in the final version.
>
> > `Weakness 2 & Question 2 & Question 3`: Analysis on reasoning depth and "Thinking Models".
>
> Thank you for this insightful suggestion. We address both aspects below:
>
> **1. Reasoning Depth (Simple vs. Multi-hop):**
> Our dataset selection was intentionally designed to cover this spectrum. Our benchmarks range from arithmetic tasks (Object Counting, GSM8K) to state-tracking multi-hop problems (Tracking Shuffled Objects, Temporal Sequences).
> As shown in Table 1, TARE and ATARE outperform prior baselines on evaluated datasets. This demonstrates that landscape smoothing delivers consistent performance gains across the spectrum, showing comparable effectiveness for both simple arithmetic and complex multi-hop reasoning tasks.
>
> **2. Performance on "Thinking Models" (New Experiments):**
> Following your suggestion to test o1-style "thinking" models, we conducted additional experiments on **DeepSeek-R1-Distill-Qwen-7B** and **DeepSeek-R1-Distill-Llama-8B**. We specifically chose these 7B/8B parameter scales to avoid the "ceiling effect" often seen with larger reasoning models on these datasets, allowing us to measure the optimization gains.
>
> **Results on Thinking Models (Acc %):**
>
> | Model | Dataset | CoT | TextGrad | Revolve | TARE | ATARE |
> | :--- | :--- | :--- | :--- | :--- | :--- | :--- |
> | **DeepSeek-R1-Distill-Qwen-7B** | Object Counting | 86.0 | 87.3 | 88.3 | 91.3 | **92.0** |
> | | Tracking Shuffled Objects | 92.5 | 94.3 | 94.0 | 96.0 | **97.0** |
> | **DeepSeek-R1-Distill-Llama-8B** | Object Counting | 94.0 | 94.0 | 94.0 | **95.0** | **95.0** |
> | | Tracking Shuffled Objects | 95.0 | 98.0 | 97.5 | **99.3** | 99.0 |
>
> **Insight:** Even for models trained with intrinsic "thinking" processes, TARE consistently boosts performance. This confirms that while these models have better reasoning capabilities, their prompt landscapes remain "sharp" and benefit from our robustness-aware optimization.

---

> ### Author Response · Authors · 2025-11-21
> **[Part 2/3] Response to Reviewer VtM1**
>
> > `Weakness 3 & Question 4`: Transferability across tasks and models.
>
> Thank you for raising this important question. We agree that transferability is key to reducing compute costs. We conducted two new sets of experiments to verify this:
>
> **1. Cross-Task Transfer (GSM8K $\rightarrow$ SVAMP):**
> We took the prompts optimized by TARE and ATARE on **GSM8K** and evaluated them directly on the **SVAMP** dataset (arithmetic word problems) without any further tuning.
>
> **Results on SVAMP (Acc %):**
>
> | Model | CoT | GSM8K-Optimized (TARE) | GSM8K-Optimized (ATARE) |
> | :--- | :--- | :--- | :--- |
> | **Llama 3.1 8B** | 85.5 | 88.0 | 88.0 |
> | **Llama 3 8B** | 78.5 | 79.0 | 80.0 |
>
> **Insight:** The prompts optimized on GSM8K transfer effectively to SVAMP, outperforming the CoT baseline. This suggests that TARE/ATARE learns robust reasoning patterns rather than overfitting to specific dataset quirks.
>
> **2. Cross-Model Transfer:**
> We tested whether prompts optimized for one model family transfer to another. We took prompts optimized for GPT-3.5(Source) and evaluated them on Llama 3 8B and Llama 3.1 8B (Target). We compare these against prompts optimized directly on the target models (Self-Opt).
>
> **Results on Target Models (Acc %):**
>
> | Task | Source Model | Target Model | TARE (Self-Opt) | TARE (Transfer) | ATARE (Self-Opt) | ATARE (Transfer) |
> | :--- | :--- | :--- | :--- | :--- | :--- | :--- |
> | **Object Counting** | GPT-3.5 | Llama 3.1 | 91.0 | 91.0 | 92.4 | 92.0 |
> | | GPT-3.5 | Llama 3 | 88.7 | 86.5 | 90.3 | 89.0 |
> | **Temporal Sequences** | GPT-3.5 | Llama 3.1 | 90.0 | 89.5 | 91.0 | 91.0 |
> | | GPT-3.5 | Llama 3 | 85.0 | 86.0 | 88.7 | 87.5 |
>
> **Insight:** The transferred prompts perform also very well. As shown in the table, the "Transfer" performance closely matches the "Self-Opt" performance (where the prompt is optimized directly on the target model). This confirms that the "flat minima" found by our method are largely model-agnostic.
>
> > `Weakness 4 & Question 5`: Optimizer choices.
>
> Thank you for this question regarding optimizer selection. We address the two aspects below:
>
> **1. Universal vs. Matched Optimizers:**
> Our results in **Table 1** suggest that the optimizer choice is largely model-agnostic. Both GPT-4o and Claude 3.5 Sonnet consistently achieved top-tier performance across all diverse target models (Llama, Gemini, GPT-3.5). We observed no distinct pattern indicating that an optimizer performs better when matched with a target model from the same family (e.g., Claude optimizing Gemini vs. Llama).
>
> **2. Target Model as Optimizer:**
> We explored this scenario in our **Resilience Analysis (Figure 4)**, where we replaced the powerful backbone with the weaker **Llama 3.1 8B** (effectively making the target model optimize itself).
> * **Result:** Performance dropped compared to using the stronger GPT-4o/Claude backbones.
> * **Reason:** Effective optimization relies on the optimizer's ability to critique the prompt and propose high-quality edits. If the optimizer has the same limitations as the target model, it often struggles to identify and correct the "sharpness" or errors that the target produces. Therefore, using a backbone with stronger reasoning capabilities than the target is generally necessary for best results.

---

> ### Author Response · Authors · 2025-11-21
> **[Part 3/3] Response to Reviewer VtM1**
>
> > `Weakness 5`: Semantic neighborhood definition.
>
> We appreciate this insightful question. In the discrete prompt space, we do not use a numeric scalar for distance. Instead, we operationalize the semantic neighborhood ($d_{text}$) through **natural language constraints** within the Generator Oracle (detailed in Appendix E.1 and E.2):
>
> * **Generator Constraints as Radius:** The "radius" $\rho$ is effectively enforced by strict instructions to maintain the core intent and theme. For ATARE, this is tightened by the sensitivity hierarchy: high-sensitivity components (Tier 1) are restricted to "minimal, subtle edits", acting as a rigid boundary to prevent semantic drift.
> * **Adversarial Exploration:** To prevent trivial or overly conservative perturbations, we explicitly instruct the generator to "explore vulnerabilities" and find variations that are potentially "worse-performing". This adversarial objective forces the model to stretch linguistic forms to uncover sharpness. Furthermore, ATARE prevents conservatism by applying stronger perturbations to low-sensitivity components (Tier 3), allowing **"creative and diverse changes"** to ensure the search covers a wider semantic range.
>
> **Revision Plan:** We will update Section 3 to map these formal neighborhood definitions ($d_{text}$) to their corresponding prompt-based constraints in the Appendix, clarifying how the generator balances semantic fidelity with adversarial exploration.
>
> > `Question 6`: Computational Cost.
>
> Thank you for this practical question. We measured the average wall-clock time (per iteration) and total API calls for a full optimization run on the **Object Counting** task.
>
> **Efficiency Comparison:**
>
> | Method | Avg. Time per Iteration (s) | Total API Calls |
> | :--- | :--- | :--- |
> | TextGrad | 104.5 | 210 |
> | Revolve | 93.8 | 210 |
> | TARE | 128.3 | 230 |
> | ATARE | 133.3 | 232 |
>
> **Analysis:** As the table shows, TARE and ATARE introduce a moderate increase in computational cost compared to TextGrad and Revolve. Considering that prompt optimization is a one-time offline process, we believe this investment is well justified to secure the significant gains in robustness and reliability during deployment.
>
> > `Question 7`: Failure Cases.
>
> Thank you for this constructive question. We identified a key scenario where TARE might struggle (Structurally Fragile Prompts): TARE relies on the assumption that a prompt has a valid "neighborhood" of variants. However, some tasks require rigid syntax or exact keywords (e.g., strict code formats or "magic words") where *any* perturbation breaks functionality. In these cases, even minor edits might push the candidate outside the valid semantic scope. TARE could misinterpret this breakage as "sharpness" (poor robustness) and attempt to optimize away from a solution that was actually correct but simply intolerant to change.

---

> ### Author Response · Authors · 2025-11-25
> **Thank you & Looking forward to further discussion**
>
> Dear Reviewer VtM1,
>
> We sincerely appreciate your commitment to the author-reviewer discussion process. To promote a clearer understanding of our rebuttal and revisions, we have summarized our responses to your key concerns below:
>
> 1) **Is there a conflict between diversity and robustness?** `Weakness 1, Question 1` We respectfully clarify that TARE enables "safe creativity" by ensuring stability in output quality rather than token determinism, thereby supporting diversity.
> 2) **How does TARE perform on "Thinking Models"?** `Weakness 2, Questions 2 & 3` We conducted additional experiments on DeepSeek-R1 models and varying reasoning depths, confirming that TARE consistently enhances performance in complex reasoning tasks.
> 3) **Does TARE demonstrate transferability?** `Weakness 3, Question 4` We empirically validated strong transferability in both cross-task (GSM8K $\rightarrow$ SVAMP) and cross-model scenarios.
> 4) **Clarifications on implementation and analysis** `Weakness 4 & 5, Questions 5, 6 & 7` We have provided detailed explanations regarding optimizer selection, semantic neighborhood definitions, computational costs, and potential failure cases.
>
> For other issues not mentioned here, please refer to our detailed rebuttal response. We truly hope these responses resolve your doubts! We respectfully look forward to further discussion with you.
>
> Warm regards,
>
> Authors

---

> ### Author Response · Authors · 2025-11-27
> **Manuscript Revision Deadline Approaching; We Are Sincerely At Your Service**
>
> Dear Reviewer VtM1,
>
> We sincerely appreciate your insightful review and your dedication to this process. Thanks to your feedback, the quality of our manuscript has been significantly improved. Given the approaching deadline, we are now submitting the revised manuscript with changes highlighted.
>
> **We also respectfully express our readiness to address any further concerns you may have.** We deeply value your insights and aim to make the best use of this revision opportunity.
>
> Thank you again for your support.
>
> Warm regards,
>
> Authors

---

### Author Response · Authors · 2025-11-27
**General Response**

Dear Reviewers,

Thank you for your thorough and insightful reviews. We sincerely appreciate your feedback, which has significantly enhanced our paper! Below, we summarize the key concerns raised and our corresponding responses:

1) **Clarification of Methodology and Notation** `Reviewers wFGW, 1hhU, v8Q2, VtM1`
We have improved the readability of the methodology section by simplifying mathematical notations, adding a Notation Table, and streamlining Figure 2. We also clarified technical details regarding the LLM-based semantic neighborhood definition and the implementation of Generator/Optimizer oracles.

2) **New Experiments and Baselines** `Reviewers wFGW, VtM1`
We have conducted additional experiments to include evolutionary baselines (APE, APO, EvoPrompt) and evaluated TARE on "Thinking Models" (DeepSeek-R1 series), demonstrating consistent performance gains.

3) **Generalization and Extensibility** `Reviewers VtM1, v8Q2`
We have empirically validated the transferability of TARE across different tasks (GSM8K to SVAMP) and model families. Furthermore, we demonstrated the extensibility of our framework to Retrieval-Augmented Generation (RAG) settings (Natural Questions, TriviaQA).

4) **Theoretical Discussions (Diversity & Validity)** `Reviewers VtM1, 1hhU`
We have elaborated on the synergy between robustness and diversity ("safe creativity") and provided post-hoc analysis (NLI checks) to verify the semantic validity of our sampling bounds.

Thank you again for your valuable feedback. We remain eager to address any further questions or concerns you may have.

Sincerely,

Authors

---

### Author Response · Authors · 2025-12-03
**Summary of Rebuttal and Revision**

Dear Area Chair,

We sincerely appreciate the time and effort you have dedicated to managing the review process for our submission. To assist with your final assessment, we wanted to provide a brief summary of the rebuttal outcomes and the updates made to the manuscript.

**Positive Reception:**
We are pleased that three of the four reviewers (VtM1, v8Q2, 1hhU) gave positive initial ratings (6, 6, 6). They consistently highlighted the novelty of our "textual sharpness" formulation and its practical value. During the discussion, we successfully addressed their specific technical inquiries:
* **Generalization & Extensibility:** We validated transferability across tasks (GSM8K $\to$ SVAMP) and model families. Furthermore, we demonstrated the extensibility of our framework to **RAG settings** (Natural Questions, TriviaQA), confirming its utility beyond standard prompting.
* **New Architectures:** We verified the method's effectiveness on **"Thinking Models"** (DeepSeek-R1 series), showing consistent performance gains even on models with intrinsic reasoning capabilities.
* **Theoretical Clarity:** We clarified the relationship between robustness and diversity **("safe creativity")** and provided post-hoc analysis (NLI checks) to verify the semantic validity of our sampling bounds.

**Addressing Reviewer wFGW:**
While Reviewer wFGW assigned a score of 4, the reviewer explicitly acknowledged the method as "reasonable" and the formulation as "concise and elegant." The reviewer's main concerns regarding baselines and clarity have been fully addressed in the revision:
* **Baselines:** We conducted the requested experiments with APE, APO, and EvoPrompt; our method consistently outperforms these evolutionary baselines.
* **Presentation & Structure:** We streamlined the main figure (Figure 2), simplified the mathematical notation, and added a Notation Table. Additionally, we moved the Related Work section to the main text as requested. We also clarified technical details regarding the LLM-based semantic neighborhood definition and the implementation of Generator/Optimizer oracles.

We believe these revisions have materially strengthened the paper and resolved the reviewers' concerns. We hope this summary is helpful for your decision.

Best regards,

Authors

---

### Meta-Review · Area_Chair_1c8B · 2026-01-06

**Summary:**

This paper proposes TARE and ATARE, frameworks for robust prompt optimization in large language models by minimizing textual sharpness, the sensitivity of prompt performance to semantically preserving paraphrases. The approach is novel, model-agnostic, and applicable in black-box settings. The empirical results are strong, with consistent gains over several baselines across reasoning tasks and model families.

Three reviewers recommended acceptance, highlighting the conceptual novelty, practical relevance, and thorough experimentation. One reviewer raised concerns about missing baselines and presentation clarity. The authors responded comprehensively, adding new comparisons, improving exposition, and extending results to RAG and thinking models. Methodological details were clarified, and the revised version is significantly stronger.

Overall, this is a solid and timely contribution to prompt optimization. I recommend acceptance.

**Reviewer Concerns:**

Reviewer wFGW raised concerns about missing baselines, lack of a Related Work section, dense notation, and unclear implementation details. The authors responded by including new baselines, moving Related Work into the main text, simplifying the methodology section, and providing detailed clarifications.

**Reviewer Scores:**

Reviewer wFGW: The author responses substantially addressed the reviewer’s concerns. If discussion were allowed, this score might have increased.

---

### Decision · Program_Chairs · 2026-01-26

Accept (Poster)